# Propagation Alone is Enough for Graph Contrastive Learning

## Abstract

Graph contrastive learning has recently gained substantial attention, leading to the development of various methodologies. In this work, we reveal that a simple training-free propagation method PROP achieves competitive results over dedicatedly designed GCL methods across a diverse set of node classification benchmarks. We elucidate the underlying rationale for PROP's effectiveness by drawing connections between the propagation operator and established unsupervised learning algorithms. To investigate the reasons for the suboptimal performance of GCL, we decouple the propagation and transformation phases of graph neural networks. Our findings indicate that existing GCL methods inadequately learns effective transformation weights while exhibiting potential for solid propagation learning. In light of these insights, we enhance PROP with learnable propagation, introducing a novel GCL method termed PROPGCL. The effectiveness of PROPGCL is demonstrated through comprehensive evaluations on node classification tasks.

## 1 Introduction

Graph contrastive learning (GCL) has emerged as a promising paradigm for learning graph representations in the unsupervised manner. By leveraging the inherent structural information in graphs, GCL has achieved state-of-the-art performance on graph learning tasks (Velickovic et al., 2019; Zhang & Chen, 2018; You et al., 2020). However, the increasing complexity of these methods, often involving intricate transformation layers, augmentation strategies, and large-scale parameter tuning, has raised questions about the necessity of such complexity for effective learning.

In this work, we challenge the conventional wisdom that highly parameterized models are essential for achieving strong performance in GCL. Instead, we explore a simple yet powerful alternative: uniform propagation, abbreviated as **PROP**, which involves no trainable layers. Remarkably, PROP demonstrates competitive performance on various node classification benchmarks, often matching or surpassing more sophisticated GCLs. This raises an important question:

*How can the simple approach perform so well compared to complex GCL methods?*

To address this, we provide a theoretical analysis of PROP, positioning it as a non-parametric method aligned with traditional unsupervised learning algorithms through iterative optimization. Additionally, we demonstrate that propagation inherently performs contrastive learning by aligning neighboring node representations, which elucidates the core strengths of PROP in enhancing feature clustering. This analysis not only demystifies the success of PROP but also highlights the potential of simpler models in graph self-supervised learning.

On the other hand, we seek to explore why GCL occasionally exhibits suboptimal performance. By adopting a decoupling perspective, we isolate and independently analyze the transformation and propagation phases within the GCL encoder. Our extensive analysis reveals a significant limitation in the transformation phase: existing GCL methods often struggle to learn meaningful transformation weights, which perform no better than random counterparts. However, the propagation phase tells a different story. We demonstrate that GCL can consistently learn informative propagation coefficients, effectively capturing structural information. This highlights the potential for developing more efficient GCL methods by prioritizing propagation over transformation.

Building on these insights, we propose a novel method, **PROPGCL**, which leverages the strengths of PROP while addressing its limitations of uniform propagation. Specifically, PROPGCL enhances PROP by learning propagation coefficients through GCL. To validate the effectiveness of PROPGCL,

we conduct experiments across a wide range of node classification benchmarks, including both homophilic and heterophilic datasets. Our results demonstrate that PROPGCL consistently outperforms existing GCL methods, and requires far fewer computational resources.

This work makes several key contributions to the field of graph contrastive learning:

- We establish PROP, a training-free method, as a strong baseline in graph self-supervised learning on node classification. We provide a theoretical framework that connects PROP to classical unsupervised learning algorithms, offering a deeper understanding of effectiveness.

- From a decoupling perspective, we reveal that existing GCL methods struggle to learn effective transformation weights while excelling at learning propagation coefficients, suggesting opportunities for efficient GCL methods by prioritizing propagation over transformation.

- We propose PROPGCL, a simple but effective method that enhances PROP by learning propagation coefficients through GCL. We rigorously evaluate PROPGCL across diverse node classification benchmarks, demonstrating its superiority over current GCL methods in terms of both accuracy and efficiency, particularly on heterophilic datasets.

## 2 RELATED WORKS

**GCL Designing Principles.** Popular GCL design approaches predominantly focus on three aspects: augmentation generation, view selection, and contrastive objectives. Augmentation strategies have been explored to enhance representation learning, such as topology-based, label-invariant, and spectral augmentations (Zhu et al., 2021b; Li et al., 2022b; Trivedi et al., 2022; Liu et al., 2022). For view selection, Guo et al. (2023b) question the necessity of positive pairs, while others focus on hard negative mining (Robinson et al., 2021; Yang et al., 2023; Niu et al., 2024). Meanwhile, contrastive objectives are often grounded in the mutual information maximization principle (Velickovic et al., 2019) or the information bottleneck principle (Xu et al., 2021). However, a critical aspect of GCL, the encoder design, has been largely overlooked, with most approaches defaulting to GCNs without thorough evaluation. In this work, we challenge this convention by decoupling the transformation and propagation phases, demonstrating that propagation alone is sufficient for effective GCL.

**Simplifying GCL Architectures.** Recent efforts in simplifying GCL have introduced various strategies aimed at reducing the complexity of existing methods. Some approaches remove the traditional augmentation process by employing K-means clustering, adding noise to the embedding space, or introducing invariant-discriminative losses (Yu et al., 2022; Lee et al., 2022; Li et al., 2023a). Zheng et al. (2022) simplify similarity computations by directly discriminating between two groups of summarized node instances, rather than comparing all nodes. Additionally, Li et al. (2023b) observe lower layers in deep networks suffer from degradation and propose an efficient blockwise training strategy. Other works explore using simpler models like MLPs or linear layers as the backbone for GCL (Liu et al., 2023; Salha et al., 2019). However, these methods continue to rely on transformation layers that introduce additional parameters. In contrast, our method eliminates transformation layers entirely, relying solely on a minimal-parameter propagation layer. This design reduces complexity while maintaining plug-and-play adaptability across various GCL frameworks.

## 3 BACKGROUND

### 3.1 GRAPH CONTRASTIVE LEARNING PIPELINES

GCL pipelines often include two stages, pretraining and evaluation. In the pretraining stage, augmented views are generated through learnable or artificial approaches and then embedded into representations via an encoder. GCL learns the encoder weights by maximizing the representation consistency between different views. The purpose of pre-training is to learn high-quality node or graph-level representations without relying on labeled data. In the evaluation stage, linear probing is commonly adopted, where a simple linear classifier is trained in a supervised manner to map the pretrained representations to the downstream label space. This enables a fair comparison of the quality of representations learned by different GCL methods.

## 3.2 Graph Convolutional Neural Networks

Graph convolutional neural Networks (GCNNs) are neural networks based on graph convolution. One of the foundational works is GCN (Kipf & Welling, 2017) which propagates information from local neighborhoods and then transforms the aggregated representation in each layer by $\mathbf{H}^{(l+1)} = \sigma(\tilde{\mathbf{A}}\mathbf{H}^{(l)}\mathbf{W}^{(l)})$, where $\mathbf{H}^{(0)} = \mathbf{X}$ denotes node features, $\tilde{\mathbf{A}}$ is the normalized adjacency matrix, $\mathbf{W}^{(l)}$ is transformation weights in the $l$-th layer, and $\sigma$ is the activation function.

**Decoupled GNNs.** In GCN, propagating information and transforming representation are inherently intertwined in each layer. However, this tight coupling of operations can lead to limitations including oversmoothing and scalability issues (Wu et al., 2019; Liu et al., 2020; Dong et al., 2021). Therefore, simpler yet effective models are proposed by decoupling the two operations (Wu et al., 2019; Gasteiger et al., 2019a; He et al., 2020). For instance, SGC (Wu et al., 2019) composes two decoupled stages of 1) *propagation* which uniformly aggregates information from $K$-hops neighboring nodes by $\mathbf{H}' = \mathbf{A}^K\mathbf{X}$, and 2) *transformation* which transforms features by $\mathbf{H} = \sigma(\mathbf{H}'\mathbf{W})$.

**Polynomial GNNs.** Despite the simplicity of SGC and its follow-ups, they are proven to perform as a low-pass filter (Balcilar et al., 2021; Nt & Maehara, 2019; Zhu et al., 2021a) and show limited expressiveness for solving various graph structures. To solve this, *polynomial GNNs* replace the uniform propagation by learnable combinations of polynomial basis functions to approximate arbitrary spectral filters (Chien et al., 2021; He et al., 2021; 2022). Similarly, polynomial GNNs can be expressed in a unified propagation and transformation framework,

$$\mathbf{H}_1 = \sum_{k=0}^{K-1} \theta_k g_k(\mathbf{L})\mathbf{X}, \qquad \text{(Propagation)}$$

$$\mathbf{H} = \sigma(\mathbf{H}_1\mathbf{W}), \qquad \text{(Transformation)}$$

where $\boldsymbol{\theta} \in \mathbb{R}^K$ is learnable *propagation coefficients*, $g_k(\mathbf{L})$ represents the *polynomial basis functions* applied to the graph Laplacian matrix $\mathbf{L}$, $\mathbf{W}$ is learnable *transformation weights*. Notably, the flexibility of learning spectral filters helps polynomial GNNs capture intricate structures in *heterophily graphs* where connected nodes tend to have different labels (He et al., 2021; 2022; Chien et al., 2021).

# 4 Uniform Propagation is A Strong Baseline for Unsupervised Learning

In this section, we demonstrate that even without trainable transformation networks, the uniform propagation is in itself a strong baseline for graph self-supervised learning (GSSL) on node classification. We reveal the rationale by connecting propagation to well-known unsupervised learning algorithms and benchmarking its performance on a wide range of homophilic and heterophilic graphs. The proofs of theorems are shown in Appendix P.

## 4.1 Propagation: A Non-parametric Learning Approach on Graph

**Propagation as nonparametric unsupervised learning.** It is widely acknowledged that propagation *alone* can provide better clustering of input features such that they are more linearly separable for node classification tasks (Kipf & Welling, 2017; Wu et al., 2019). By aggregating features from neighboring nodes, cascaded propagation operators perform iterative updates of node features,

$$\mathbf{H}^{(k+1)} = \hat{\mathbf{A}}\mathbf{H}^{(k)}, \qquad (1)$$

where $\mathbf{H}^{(0)} = \mathbf{X}$ is node features, $\hat{\mathbf{A}} = \mathbf{D}^{-\frac{1}{2}}\mathbf{A}\mathbf{D}^{-\frac{1}{2}}$ is normalized adjacency matrix, and $k$ indexes the propagation step. The following theorem shows that with an appropriate learning step, the propagation process realizes the gradient descent of the Dirichlet energy, which measures the feature distance between neighboring nodes (Zhu et al., 2021a).

**Theorem 4.1.** *For a learning step size of $\alpha = 0.5$, the propagation procedure of Equation 1 optimizes the following Dirichlet energy objective and converges to a state where the energy $\mathcal{L}(\mathbf{H}^{(K)}) \to 0$ as $K \to +\infty$ for non-bipartite graphs.*

$$\mathcal{L}(H) = \mathbf{H}^\top \hat{\mathbf{L}}\mathbf{H} = \sum_{i,j} \hat{\mathbf{A}}_{ij}\|\mathbf{H}_i - \mathbf{H}_j\|^2, \qquad (2)$$

In this way, propagation alone can be regarded as a *non-parametric* approach to unsupervised learning based on *iterative optimization*, similar to k-means and compressed sensing (Shehu et al., 2020).

**Propagation as graph contrastive learning.** In fact, the propagation operator can also be understood as a special GCL method, where the positive samples are randomly drawn from *neighboring* nodes. Define the joint distribution of positive pairs as $p(x_i, x_j) = A_{ij} / \sum_{i,j} A_{ij}$, where $A_{ij}$ denotes the normalized edge weight in the adjacency matrix. The alignment loss between positive pairs becomes,

$$\mathcal{L}_{\text{align}}(f) = -\mathbb{E}_{x_i, x_j \sim p(x_i, x_j)}[f(x_i)^\top f(x_j)]. \tag{3}$$

Intuitively, this alignment task will bring the representation of neighboring nodes together. In fact, as shown in the following theorem, propagation minimizes this alignment loss at its optimum, indicating that as an architecture component, the propagation can perform implicitly (though not exactly is) contrastive learning.

**Theorem 4.2.** *Let $f_k(x_i) = \mathbf{H}_i^{(k)}, \forall\, i \in [N]$ be unit vectors, then $\lim_{k \to \infty} \mathcal{L}_{\text{align}}(f_k) = -1$.*

The connection between PROP and unsupervised learning provides insight into why propagation alone can deliver competitive performance in unsupervised settings. Notably, our analysis is not on the full GCN framework with trainable weights, but on the propagation operation only. Indeed, GCN's transformation stage cannot be equated with some form of contrastive learning, evidenced by the poor expressive power of untrained GCNs in unsupervised settings (Suresh et al., 2021).

## 4.2 Benchmark Propagation among Unsupervised Node Classification Baselines

We compare the uniform propagation operation with representative GSSL methods in a unified setting on homophily and heterophily benchmarks. Experiments show that PROP is highly competitive among GSSL baselines. Detailed experimental details are shown in Appendix O.

**Method.** The connections above reveal that iterative propagation can be understood as a special non-parametric unsupervised learning algorithm. We denote the propagation-only operation as **PROP**, which aggregates features within $K$-hop neighbors without any trainable weights, *i.e.*,

$$\mathbf{H}_{\text{PROP}} = \tilde{\mathbf{A}}^K \mathbf{X}, \tag{4}$$

where $\tilde{\mathbf{A}}$ is the same propagation matrix used in GCN, *i.e.*, $\tilde{\mathbf{A}} = \mathbf{D}'^{-\frac{1}{2}} \mathbf{A}' \mathbf{D}'^{-\frac{1}{2}}$ with $\mathbf{A}' = \mathbf{A} + \mathbf{I}$. Note that the formulation of PROP has no essential difference from SGC. The main focus of this work is not to propose a new alternative method as "PROP", just to have a critical understanding of existing GCL methods through an ablated baseline. The naming as "PROP" instead of "SGC" is to make this point clear and avoid confusion with the common use of SGC in GCL literature, which contains weights $\mathbf{W}$ in practice (Chen & Kou, 2023; Gao et al., 2023).

**Datasets.** For homophily benchmarks, we choose popular citation network datasets Cora, CiteSeer, and PubMed (Sen et al., 2008; Namata et al., 2012), Amazon co-purchase datasets Photo, Computers (Shchur et al., 2018). For heterophily benchmarks, we include Wikipedia datasets Squirrel, Chameleon (Rozemberczki et al., 2021) and WebKB datasets Texas, Wisconsin, and Cornell (Pei et al., 2020).

**Baselines.** We consider two categories of representative GSSL methods as baselines: 1) traditional graph embeddings DeepWalk (Perozzi et al., 2014) and Node2Vec (Grover & Leskovec, 2016), 2) deep learning methods including graph autoencoders GAE (Kipf & Welling, 2016), VGAE (Kipf & Welling, 2016), and contrastive learning methods GRACE (Zhu et al., 2020b), DGI (Velickovic et al., 2019), GCA (Zhu et al., 2021c), MVGRL (Hassani & Khasahmadi, 2020), ProGCL (Xia et al., 2022), CCA-SSG (Zhang et al., 2021), BGRL (Thakoor et al., 2022). Given the superiority of polynomial GNNs, we also compare replacing the vanilla GCN encoder in GCLs with polynomial GNNs.

**Settings.** Following Zhu et al. (2020b); Hassani & Khasahmadi (2020), we use the linear evaluation protocol, where the model is trained unsupervised and the learned representations are fed into a linear logistic regression classifier. We follow Chien et al. (2021); Chen et al. (2024) to randomly split the nodes into 60%, 20%, and 20%. We also conduct fixed-splitting experiments in Appendix E.

**Results.** We show the experimental results in Table 1. **Even without computationally expensive training, PROP maintains a superior performance over competing methods.** For homophily benchmarks, PROP achieves comparable performances with other GSSL methods. For heterophilic benchmarks, PROP exceeds other methods by a large margin, including GCLs with polynomial GNNs. For example, PROP achieves 58.5% on Squirrel while the runner-up only has 49.5% accuracy.

Notably, GSSL baselines require time-intensive training and extensive hyperparameter tuning, while **training-free PROP operates without backpropagation and has only one hyperparameter, the propagation step** $K$. This efficiency highlights the strength of PROP. In Appendix G, we further present the accuracy trends of PROP across different propagation steps.

Table 1: Test accuracy (%) of PROP and other graph self-supervised methods on node classification benchmarks. Each experiment is repeated ten times with mean and standard derivation of accuracy score. Red indicates the best method, while blue represents the second-best choice.

| Training | Encoder | Homophily | | | | | | Heterophily | | | | | |
|---|---|---|---|---|---|---|---|---|---|---|---|---|---|
| | | Cora | CiteSeer | PubMed | Computers | Photo | **Mean** | Squirrel | Chameleon | Texas | Wisconsin | Cornell | **Mean** |
| Supervised | GCN | 87.5 ± 1.0 | 80.2 ± 0.6 | 87.0 ± 0.3 | 88.4 ± 0.3 | 93.5 ± 0.4 | 87.3 | 47.6 ± 0.8 | 64.1 ± 1.6 | 76.4 ± 4.1 | 62.6 ± 2.8 | 64.4 ± 4.1 | 63.0 |
| | ChebNetII | 87.2 ± 0.8 | 79.9 ± 0.8 | 88.5 ± 0.1 | 90.1 ± 0.3 | 94.9 ± 0.3 | 88.1 | 56.7 ± 1.3 | 72.3 ± 1.5 | 92.6 ± 1.8 | 89.3 ± 3.6 | 90.5 ± 1.6 | 80.3 |
| *Unsupervised Graph Embedding* | | | | | | | | | | | | | |
| DeepWalk | Word2Vec | 80.6 ± 0.8 | 63.1 ± 1.0 | 81.9 ± 0.2 | 87.3 ± 0.4 | 91.5 ± 0.5 | 80.9 | 43.3 ± 0.7 | 60.8 ± 1.3 | 53.4 ± 4.8 | 43.6 ± 4.1 | 44.6 ± 3.1 | 49.2 |
| Node2Vec | Word2Vec | 80.2 ± 1.2 | 68.1 ± 0.9 | 80.7 ± 0.3 | 85.5 ± 0.4 | 90.3 ± 0.5 | 81.0 | 39.7 ± 1.0 | 59.2 ± 1.1 | 56.2 ± 4.6 | 43.6 ± 2.8 | 45.6 ± 2.8 | 48.9 |
| *GSSL with Vanilla GNNs* | | | | | | | | | | | | | |
| GRACE | GCN | 86.9 ± 1.0 | 75.6 ± 0.7 | 85.3 ± 0.2 | 82.3 ± 0.2 | 90.1 ± 0.3 | 84.0 | 43.8 ± 1.0 | 62.3 ± 0.9 | 73.6 ± 4.3 | 67.0 ± 1.8 | 65.6 ± 9.0 | 62.5 |
| DGI | GCN | 85.8 ± 1.0 | 78.6 ± 0.7 | 82.3 ± 0.3 | 79.6 ± 0.4 | 80.6 ± 1.2 | 81.4 | 37.1 ± 0.8 | 52.4 ± 1.3 | 82.6 ± 2.3 | 72.1 ± 2.4 | 80.3 ± 2.0 | 64.9 |
| GAE | GCN | 84.9 ± 1.3 | 75.7 ± 0.8 | 84.7 ± 0.3 | 76.3 ± 0.5 | 90.5 ± 0.3 | 82.4 | 36.2 ± 0.9 | 56.8 ± 1.6 | 60.0 ± 4.3 | 56.9 ± 4.9 | 57.0 ± 6.7 | 53.4 |
| VGAE | GCN | 85.1 ± 1.0 | 75.6 ± 0.7 | 84.6 ± 0.3 | 76.4 ± 0.5 | 88.3 ± 0.6 | 82.0 | 43.4 ± 0.6 | 61.4 ± 1.0 | 73.1 ± 3.4 | 60.8 ± 4.5 | 65.0 ± 7.4 | 60.8 |
| MVGRL | GCN | 84.0 ± 1.0 | 74.5 ± 0.8 | 83.6 ± 0.4 | 83.5 ± 0.5 | 89.2 ± 0.4 | 83.0 | 31.3 ± 0.6 | 57.9 ± 1.6 | 77.7 ± 2.0 | 65.8 ± 3.5 | 67.5 ± 7.9 | 60.0 |
| CCA-SSG | GCN | 86.7 ± 0.9 | 79.7 ± 0.6 | 84.8 ± 0.4 | 82.8 ± 0.3 | 91.2 ± 0.4 | 85.0 | 40.6 ± 0.7 | 57.8 ± 1.0 | 79.3 ± 3.1 | 71.1 ± 1.4 | 72.6 ± 4.9 | 64.3 |
| BGRL | GCN | 85.1 ± 0.7 | 76.5 ± 0.9 | 84.0 ± 0.2 | 82.8 ± 0.4 | 86.1 ± 0.4 | 82.9 | 36.8 ± 0.7 | 55.5 ± 1.8 | 79.7 ± 3.6 | 67.5 ± 3.9 | 71.0 ± 10.3 | 62.1 |
| GCA | GCN | 84.7 ± 1.0 | 76.5 ± 0.8 | 85.0 ± 0.2 | 79.3 ± 0.2 | 89.5 ± 0.3 | 83.0 | 41.0 ± 0.9 | 59.4 ± 1.1 | 78.0 ± 2.6 | 74.0 ± 2.1 | 66.9 ± 7.1 | 63.8 |
| ProGCL | GCN | 84.6 ± 1.0 | 78.0 ± 0.5 | 86.9 ± 0.2 | 91.2 ± 0.5 | 84.3 ± 0.4 | 85.0 | 49.5 ± 0.6 | 67.5 ± 1.1 | 77.9 ± 3.8 | 71.4 ± 2.5 | 66.6 ± 11.3 | 66.6 |
| *GSSL with Polynomial GNNs* | | | | | | | | | | | | | |
| | ChebNetII | 83.4 ± 0.9 | 74.8 ± 0.6 | 84.9 ± 0.3 | 84.1 ± 0.4 | 89.2 ± 0.5 | 83.3 | 37.9 ± 0.8 | 55.7 ± 1.0 | 77.9 ± 2.8 | 86.4 ± 3.6 | 75.7 ± 3.6 | 66.7 |
| GRACE | BernNet | 82.8 ± 1.1 | 75.4 ± 0.9 | 84.2 ± 0.2 | 85.8 ± 0.4 | 89.7 ± 0.4 | 83.6 | 40.6 ± 0.7 | 54.7 ± 1.3 | 75.4 ± 3.6 | 88.3 ± 3.1 | 74.2 ± 4.1 | 66.7 |
| | GPRGNN | 82.4 ± 1.0 | 75.4 ± 1.0 | 84.6 ± 0.3 | 81.0 ± 0.7 | 90.1 ± 0.5 | 82.7 | 38.2 ± 0.7 | 53.8 ± 1.4 | 78.7 ± 4.4 | 71.3 ± 3.9 | 77.7 ± 5.7 | 63.9 |
| | ChebNetII | 83.4 ± 0.9 | 71.3 ± 1.2 | 81.9 ± 0.4 | 79.6 ± 0.3 | 78.7 ± 0.7 | 79.0 | 34.3 ± 0.6 | 51.0 ± 1.0 | 80.8 ± 2.1 | 81.8 ± 3.0 | 80.8 ± 1.6 | 65.7 |
| DGI | BernNet | 81.5 ± 1.0 | 73.4 ± 0.5 | 82.8 ± 0.2 | 79.2 ± 0.6 | 78.3 ± 0.5 | 79.1 | 32.4 ± 0.9 | 47.4 ± 1.8 | 82.8 ± 2.1 | 78.3 ± 2.3 | 83.6 ± 2.6 | 64.9 |
| | GPRGNN | 82.4 ± 1.4 | 74.7 ± 1.0 | 80.9 ± 0.2 | 77.8 ± 0.6 | 77.8 ± 0.6 | 78.1 | 32.8 ± 0.6 | 51.0 ± 1.4 | 80.0 ± 2.0 | 70.0 ± 3.8 | 78.9 ± 3.8 | 62.5 |
| *Training-free Method* | | | | | | | | | | | | | |
| \ | **PROP** | 85.5 ± 0.8 | 78.9 ± 0.6 | 82.9 ± 0.5 | 87.5 ± 0.5 | 93.0 ± 0.3 | 85.6 | 58.5 ± 1.0 | 68.8 ± 1.4 | 86.2 ± 3.1 | 89.0 ± 3.3 | 86.2 ± 3.1 | 77.8 |

## 5 DISSECTING THE LIMITATIONS OF GNNs IN GCL

The preceding experiments reveal that existing GCL methods perform worse than the simple PROP. In this section, we seek to understand the rationale behind this. For this aim, we analyze the decoupling of the propagation and transformation phases, a widely adopted perspective in GNNs designing (Gasteiger et al., 2019a;b; Li et al., 2022a) and scalability considerations (Yu et al., 2024; Liao et al., 2024). Through this analytical framework, we aim to identify which phase is inadequately learned within the context of GCL.

### 5.1 FEATURE TRANSFORMATION IS INEFFECTIVE IN GCL

To determine whether GCL effectively learns *transformation* weights, we consider a decoupled encoder, *i.e.*, $\mathbf{H}_{\text{PROP}} = \hat{\mathbf{A}}^K \mathbf{X}$ followed by two transformation layers $\mathbf{H} = \sigma(\mathbf{H}_{\text{PROP}} \mathbf{W}_1) \mathbf{W}_2$ where $\mathbf{W}_1$ and $\mathbf{W}_2$ are the transformation weights. The unweighted propagation enables only focusing on the transformation weights.

The core idea is comparing the transformation weights learned in GCL with random matrices. In practice, we first train the transformation weights through GCL methods. Then we replace the learned transformation weights with a random matrix whose element is independently sampled from a Gaussian distribution $\mathcal{N}(\mu, \sigma)$, where $\mu$ is the mean and $\sigma$ is the standard derivation. Representations generated by the randomized model are then fed into the downstream task for evaluation.

As shown in Table 2, **the transformation weights learned by GCL are no better than random**. The model with random weights $\mathbf{W}_1$ and $\mathbf{W}_2$ attains a performance of 71.42%, remarkably close to the 71.76% reached by the transformation weights learned through GCL. Notably, while *random projection* (Bingham & Mannila, 2001) is well-established in the literature and proven effective in various works (Bauw et al., 2021; Li et al., 2006; Freund et al., 2007), GCL should aim to *learn*

weights tailored on data, rather than relying on a random matrix. Therefore, the results indicate that GCL fails to learn informative transformation weights as expected. We hypothesize the failure stems from the unsupervised nature of the task, which leads to inefficient optimization in the absence of sufficient guidance.

Empirically, we compare the difference between the transformation weights learned by supervised learning (SL) and GCL. Figure 1(a) and Figure 1(b) illustrate the heatmaps and distributions of the transformation weights learned in SL and GCL. The SL weights have a substantial variance across different neuron positions as revealed in the heatmap, and the distribution exhibits a leptokurtic-like shape [1]. However, the GCL weights exhibit more uniform smoothing and closely resemble a normal distribution, aligning with the randomization experiments discussed earlier. These observations suggest that specific neurons in SL play pivotal roles in distinguishing features, whereas the GCL learning process appears overly generalized, diminishing the richness of feature representation.

Table 2: Test accuracy (%) of node classification benchmarks, comparing the transformation weights ($\mathbf{W}_1$ and/or $\mathbf{W}_2$) learned through GCL with random weights. We present the GRACE method for space limit and results of other GCL methods are shown in Appendix D. Red indicates the best method, while underlined represents the second-best choice.

| Training | Cora | CiteSeer | PubMed | Squirrel | Chameleon | Texas | Wisconsin | Cornell | **Mean** |
|---|---|---|---|---|---|---|---|---|---|
| GCL | **83.32 ± 1.00** | **73.02 ± 0.78** | 82.63 ± 0.41 | 36.03 ± 0.66 | 59.04 ± 1.69 | 84.59 ± 3.11 | 74.75 ± 3.00 | **80.66 ± 1.80** | 71.76 |
| Randomize $\mathbf{W}_1$ | 79.75 ± 0.99 | 68.64 ± 0.86 | 82.65 ± 0.39 | 34.77 ± 0.67 | **61.38 ± 1.29** | 85.74 ± 3.28 | **76.25 ± 3.38** | 80.49 ± 1.97 | 71.21 |
| Randomize $\mathbf{W}_2$ | 82.38 ± 1.08 | 70.46 ± 0.97 | 82.70 ± 0.33 | 36.70 ± 0.70 | 59.39 ± 1.16 | 85.74 ± 2.95 | 73.13 ± 2.25 | 80.33 ± 1.80 | 71.35 |
| Randomize both | 81.31 ± 0.85 | 68.64 ± 1.06 | **82.74 ± 0.27** | **37.08 ± 1.17** | 61.12 ± 0.99 | **85.90 ± 1.97** | 74.25 ± 1.63 | 80.33 ± 1.81 | 71.42 |

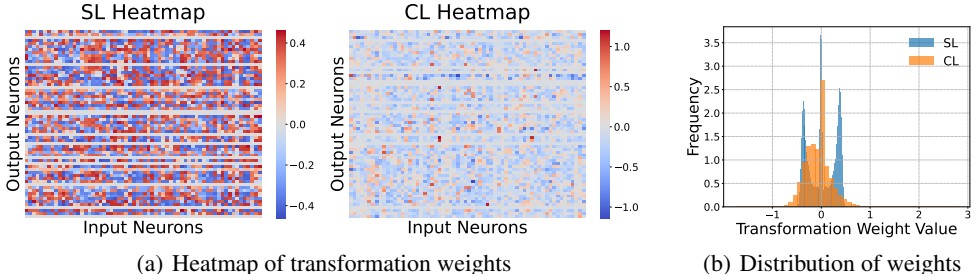

(a) Heatmap of transformation weights      (b) Distribution of weights

Figure 1: Characterization of the transformation weights learned by SL and GCL. We show an illustration of the Cora dataset using the GRACE method. Full results can be found in Appendix L.

## 5.2 LEARNING PROPAGATION IS PROMISING IN GCL

Now, we take a comprehensive view of both the transformation and propagation phases. While polynomial GNNs incorporate learnable parameters in both, GCLs utilizing polynomial GNNs, as shown in Section 4.2, tend to underperform. This issue has been recognized in prior work, often attributed to the mismatch between the strong fitting capacity of polynomial filters and the lack of supervision signals in self-supervised learning (Chen et al., 2022; 2024). However, through the following experiments, we demonstrate that GCLs are capable of learning effective filters.

From the decoupling perspective, there are three *conjectures* as to why polynomial GNNs perform poorly in GCL: 1) GCL learns suboptimal transformation weights, 2) GCL learns ineffective propagation coefficients, or 3) a combination of both. To investigate the cause, we separately replace the propagation coefficients $\theta$ and the transformation weights $\mathbf{W}$ with well-trained parameters from a supervised setting. Specifically, we first train polynomial GNNs via supervised learning and save the optimized parameters as $\mathbf{W}_{\text{SL}}$ and $\theta_{\text{SL}}$. We then proceed with the following experiments:

**Experiment 1 (Fix-propagation).** Corresponding to the first conjecture, we initialize and freeze $\theta$ with the well-trained $\theta_{\text{SL}}$, and only learn $\mathbf{W}$ through GCL. Representations are generated by the fixed propagation coefficients and learned transformation weights.

---

[1] A leptokurtic-like shape indicates a sharp concentration around the mean.

**Experiment 2 (Fix-transformation).** Corresponding to the second conjecture, we initialize and freeze $\mathbf{W}$ with the well-trained $\mathbf{W}_{\text{SL}}$, and only learn $\boldsymbol{\theta}$ through GCL. Representations are generated by the learned propagation coefficients and fixed transformation weights.

**Experiment 3 (All-one baseline).** To verify that GCL indeed learns effective propagation coefficients, we further consider a baseline with fixed well-trained transformation weights and an all-one propagation vector $\mathbb{1}$.

The experimental results are summarized in Table 3. For the first conjecture, the fix-propagation model averages 72.19%, significantly lower than the supervised model's 80.41%, and sometimes even underperforms the original GCL method. It indicates that **GCL struggles to learn effective transformation weights** (like $\mathbf{W}_{\text{SL}}$) **even with strong filters**. For the second conjecture, the fix-transformation model achieves an average performance of 79.56%, closely matching that of the supervised model. In contrast, the all-one baseline yields a lower accuracy of 75.56%, confirming that the learned propagation coefficients are effective. Thus, **GCL can learn good propagation coefficients with well-trained transformation weights**. For further validation, flip experiments replacing supervised parameters with GCL-trained ones are detailed in Appendix F, with further comparisons of learned propagation coefficients in Appendix K.

Table 3: Test accuracy (%) of node classification benchmarks. We freeze the propagation coefficients with optimal $\boldsymbol{\theta}_{\text{SL}}$ (or the transformation weights with $\mathbf{W}_{\text{SL}}$), and *learn* the transformation weights (or propagation coefficients) through GCL. $\mathbb{1}$ denotes an all-one vector. Red indicates the best, while underlined represents the second-best choice.

| | $\theta$ | $\mathbf{W}$ | Cora | CiteSeer | PubMed | Squirrel | Chameleon | Texas | Wisconsin | Cornell | **Mean** |
|---|---|---|---|---|---|---|---|---|---|---|---|
| SL | $\theta_{SL}$ | $\mathbf{W}_{SL}$ | **88.39 ± 0.74** | 79.67 ± 0.72 | 87.11 ± 0.25 | **49.34 ± 1.09** | **69.52 ± 0.96** | 89.67 ± 2.13 | 91.25 ± 2.75 | 88.36 ± 3.11 | **80.41** |
| GCL | *Learn* | *Learn* | 83.42 ± 0.92 | 74.79 ± 0.57 | 84.92 ± 0.26 | 37.90 ± 0.79 | 55.67 ± 0.96 | 77.87 ± 2.79 | 86.38 ± 3.63 | 75.74 ± 3.61 | 72.09 |
| Fix-propagation | $\theta_{SL}$ | *Learn* | 80.26 ± 0.95 | 76.15 ± 0.80 | 82.41 ± 0.64 | 40.31 ± 0.60 | 59.06 ± 1.58 | 78.69 ± 4.75 | 87.88 ± 2.75 | 72.79 ± 5.57 | 72.19 |
| Fix-transformation | *Learn* | $\mathbf{W}_{SL}$ | 87.47 ± 0.67 | **81.11 ± 0.55** | **87.69 ± 0.24** | 45.74 ± 1.57 | 64.95 ± 2.19 | **90.00 ± 2.46** | 91.38 ± 3.50 | **88.85 ± 4.10** | 79.65 |
| All-one baseline | $\mathbb{1}$ | $\mathbf{W}_{SL}$ | 78.24 ± 0.92 | 78.72 ± 0.48 | 84.75 ± 0.33 | 35.98 ± 0.77 | 59.61 ± 1.07 | 89.34 ± 3.93 | 89.38 ± 2.25 | 88.49 ± 3.77 | 75.56 |

# 6 PROPGCL: SIMPLE GRAPH CONTRASTIVE LEARNING THAT ONLY LEARNS PROPAGATION

In Section 5.2, we demonstrate that GCL can effectively learn in the propagation phase, provided well-trained transformation weights. This insight suggests potential *few-shot learning* applications, with preliminary explorations are discussed in Appendix C. However, in the unsupervised setting, optimal transformation weights are unattainable and GCL tends to learn overly smooth weights. Possible remedies include enforcing weight sparsity via $l_1$ regularization, applying whitening techniques (Bell & Sejnowski, 1997), or utilizing normalization methods (Hua et al., 2021; Guo et al., 2023a). Nevertheless, these methods fail to address the issue as reported in Appendix H.

## 6.1 PROPGCL

Fortunately, the strong performance of the training-free PROP suggests that a simple model with few trainable parameters may suffice to achieve competitive results. Inspired by findings in above sections, we propose to only learn propagation coefficients within the GCL framework. Specifically, for a given GCL backbone method, we revise it by only replacing the original encoder with the following learnable spectral propagation,

$$\mathbf{H}_{\text{PROPGCL}} = \sum_{k=0}^{K-1} \theta_k g_k(\mathbf{L})\mathbf{X}, \qquad (5)$$

where $\boldsymbol{\theta} \in \mathbb{R}^K$ is the learnable propagation coefficients, and $g_k(\mathbf{L})$ represents the polynomial basis functions. For clarity, we denote the revised backbone GCL method with the prefix *PROP*. Despite largely reducing the trainable weights, the method delivers surprisingly competitive performance as shown in the following experiments.

## 6.2 EXPERIMENTAL RESULTS

**Settings.** We keep experimental settings the same as Section 4.2. Besides the previously considered benchmarks, we also evaluate the recently proposed heterophily benchmark (Platonov et al., 2023b) and large benchmarks ogbn-arxiv (Hu et al., 2020) and ogbn-products.

**Baselines.** For the baseline, we include PROP, which outperforms well-known GSSL methods as outlined in Section 4.2. Additionally, we consider recently proposed GCL methods specifically designed for heterophilic graphs, including PolyGCL (Chen et al., 2024), HGRL (Chen et al., 2022), GraphACL (Xiao et al., 2024), SP-GCL (Wang et al., 2023), and DSSL (Xiao et al., 2022). In our approach, we choose GRACE, DGI, and a scale-friendly GGD method (Zheng et al., 2022) as backbones. For example, we replace the original GCN encoder in GRACE with the formulation in Equation 5, referring to the modified method as PROP-GRACE. We utilize the Chebyshev basis as the polynomial function and conduct ablation study of basis choices in Appendix N.

**Results.** The results on node classification benchmarks are presented from Table 4 to Table 7. **Our method surpasses the PROP baseline and GCL methods on most benchmarks.** For homophily benchmarks (Table 5), PROP-GRACE achieves the highest average accuracy of 88.76%, with PROP-DGI securing the second-highest at 88.42%. Our approach attains the best performance in 3 out of 6 benchmarks and performs comparably to the best methods in the remaining cases. On popular heterophily benchmarks (Table 6), PROP-DGI attains an average accuracy of 73.71%, surpassing the state-of-the-art PolyGCL by a margin of 4.23%, and ranks first on 4 out of 6 benchmarks. On recently proposed heterophily benchmarks (Platonov et al., 2023b) (Table 7), PROP-DGI (PROP-GRACE is excluded for the scaling of GRACE) achieves the best results in 2 out of 5 benchmarks and attains an average performance of 70.22%, second only to PolyGCL's 71.68%. Notably, PolyGCL is optimized for heterophily graphs, whereas PROP-DGI builds on the simpler DGI framework. On large benchmarks (Table 4), our method performs comparably with the corresponding backbone method. For instance, on ogbn-arxiv our PROP-DGI only falls behind DGI by 0.29% on test accuracy, but at the advantage of higher time and memory efficiency. In conclusion, PROPGCL exhibits competitive performance on diverse node classification benchmarks, especially heterophily datasets where many traditional GCL methods struggle. Moreover, thanks to removing transformation weights, PROPGCL shows a great advantage in computational and memory efficiency as seen in the following section.

Table 4: Test and validation accuracy (%) on ogbn-arxiv and ogbn-products, comparing PROPGCL with baselines.

| Benchmark | Method | Val Acc | Test Acc |
|---|---|---|---|
| ogbn-arxiv | GGD | $71.11 \pm 0.13$ | $70.26 \pm 0.15$ |
| | **PROP-GGD** | $70.78 \pm 0.07$ | $69.71 \pm 0.06$ |
| | DGI | $71.05 \pm 0.12$ | $70.09 \pm 0.12$ |
| | **PROP-DGI** | $70.82 \pm 0.01$ | $69.80 \pm 0.01$ |
| ogbn-products | GGD | $90.59 \pm 0.06$ | $75.49 \pm 0.19$ |
| | **PROP-GGD** | $87.88 \pm 0.15$ | $73.57 \pm 0.15$ |

Table 5: Test accuracy (%) of homophily node classification benchmarks, comparing PROPGCL with other baselines. Red indicates the best method, while underlined represents the second-best.

| Method | Cora | CiteSeer | PubMed | Photo | Computers | CS | Mean |
|---|---|---|---|---|---|---|---|
| PROP | $85.48 \pm 0.75$ | $78.87 \pm 0.63$ | $82.89 \pm 0.48$ | $93.01 \pm 0.28$ | $87.54 \pm 0.47$ | $95.15 \pm 0.19$ | 87.16 |
| PolyGCL | $86.19 \pm 0.76$ | $79.07 \pm 0.82$ | $\underline{86.69 \pm 0.24}$ | $92.70 \pm 0.18$ | $88.91 \pm 0.25$ | $95.30 \pm 0.07$ | 88.14 |
| SP-GCL | $84.68 \pm 0.81$ | $76.43 \pm 0.63$ | $\textbf{86.98} \pm \textbf{0.23}$ | $92.65 \pm 0.48$ | $\underline{89.04 \pm 0.35}$ | $91.95 \pm 0.24$ | 86.91 |
| HGRL | $85.39 \pm 1.00$ | $79.84 \pm 0.91$ | $85.12 \pm 0.30$ | $\textbf{93.61} \pm \textbf{0.22}$ | $85.89 \pm 0.22$ | $95.57 \pm 0.12$ | 87.57 |
| GraphACL | $87.41 \pm 1.00$ | $79.17 \pm 0.55$ | $85.71 \pm 0.27$ | $92.86 \pm 0.33$ | $86.43 \pm 0.35$ | $94.17 \pm 0.16$ | 87.63 |
| DSSL | $\textbf{87.60} \pm \textbf{1.18}$ | $79.52 \pm 1.10$ | $86.62 \pm 0.24$ | $93.15 \pm 0.46$ | $88.53 \pm 0.38$ | $94.10 \pm 0.18$ | 88.25 |
| **PROP-GRACE** | $\underline{87.42 \pm 0.95}$ | $\textbf{81.56} \pm \textbf{0.83}$ | $86.19 \pm 0.35$ | $\underline{93.32 \pm 0.31}$ | $88.12 \pm 0.23$ | $\textbf{95.95} \pm \textbf{0.14}$ | **88.76** |
| **PROP-DGI** | $86.19 \pm 1.05$ | $\underline{80.78 \pm 0.65}$ | $85.14 \pm 0.22$ | $92.78 \pm 0.37$ | $\textbf{89.81} \pm \textbf{0.20}$ | $\underline{95.82 \pm 0.18}$ | $\underline{88.42}$ |

## 7 EFFICIENCY ANALYSIS

Thanks to exclusion of transformation weights, **PROPGCL demonstrates superior efficiency compared to corresponding baseline methods in terms of both computational time and memory usage.** As shown in Table 8, PROP-GRACE saves 84.29% training time per epoch for GRACE on Coauthor CS. For memory consumption, PROP-GRACE consumes over 99% less memory in the encoder for different benchmarks than GRACE. We also conduct evaluation with different basis choices and consistently find a boost of efficiency. See Appendix M for the full table.

Table 6: Test accuracy (%) of heterophily node classification benchmarks, comparing PROPGCL and other baselines. Red indicates the best method, while underlined represents the second-best.

| Method | Squirrel | Chameleon | Actor | Texas | Wisconsin | Cornell | Mean |
|---|---|---|---|---|---|---|---|
| PROP | $\underline{58.48 \pm 1.03}$ | $68.82 \pm 1.42$ | $39.36 \pm 0.91$ | $86.23 \pm 3.11$ | $89.00 \pm 3.25$ | $86.23 \pm 3.11$ | $71.35$ |
| PolyGCL | $56.09 \pm 0.87$ | $\underline{72.17 \pm 1.12}$ | $40.50 \pm 0.78$ | $86.72 \pm 2.13$ | $85.50 \pm 4.00$ | $75.90 \pm 2.46$ | $69.48$ |
| SP-GCL | $58.11 \pm 0.70$ | $70.98 \pm 0.90$ | $30.40 \pm 1.11$ | $81.97 \pm 2.79$ | $76.00 \pm 3.75$ | $65.74 \pm 6.39$ | $63.87$ |
| HGRL | $38.89 \pm 0.85$ | $55.69 \pm 1.03$ | $37.09 \pm 0.68$ | $84.10 \pm 4.75$ | $86.13 \pm 3.00$ | $84.59 \pm 4.27$ | $64.57$ |
| GraphACL | $53.77 \pm 0.89$ | $66.94 \pm 1.05$ | $38.73 \pm 0.86$ | $84.43 \pm 1.80$ | $80.00 \pm 2.50$ | $79.51 \pm 1.80$ | $67.23$ |
| DSSL | $47.56 \pm 0.98$ | $68.85 \pm 3.77$ | $35.64 \pm 0.51$ | $85.90 \pm 2.62$ | $79.00 \pm 2.75$ | $80.98 \pm 2.13$ | $67.77$ |
| **PROP-GRACE** | $55.09 \pm 0.81$ | $71.73 \pm 1.18$ | $39.35 \pm 0.81$ | $\underline{89.84 \pm 1.81}$ | $88.50 \pm 3.63$ | $\underline{86.72 \pm 2.46}$ | $\underline{71.87}$ |
| **PROP-DGI** | $60.53 \pm 0.66$ | $74.11 \pm 0.96$ | $39.53 \pm 0.84$ | $91.80 \pm 2.30$ | $\underline{88.88 \pm 2.50}$ | $87.38 \pm 2.62$ | $73.71$ |

Table 7: Test accuracy (%) of recent heterophily node classification benchmarks, comparing PROP-DGI and baselines. Red indicates the best method, while underlined represents the second-best.

| Method | roman empire | amazon ratings | minesweeper | tolokers | questions | Mean |
|---|---|---|---|---|---|---|
| PROP | $63.95 \pm 0.33$ | $40.22 \pm 0.22$ | $74.10 \pm 0.58$ | $71.74 \pm 0.51$ | $70.23 \pm 0.59$ | $64.05$ |
| PolyGCL | $\underline{71.11 \pm 0.47}$ | $44.09 \pm 0.31$ | $86.11 \pm 0.41$ | $83.70 \pm 0.59$ | $73.41 \pm 0.84$ | $71.68$ |
| SP-GCL | $55.72 \pm 0.34$ | $43.02 \pm 0.38$ | $72.38 \pm 0.64$ | $76.69 \pm 0.60$ | $\underline{73.91 \pm 0.74}$ | $64.34$ |
| HGRL | $63.31 \pm 0.33$ | $39.65 \pm 0.32$ | $52.14 \pm 0.44$ | $74.34 \pm 0.45$ | OOM | $-$ |
| GraphACL | $59.66 \pm 0.37$ | $42.68 \pm 0.19$ | $67.73 \pm 0.72$ | $74.93 \pm 0.73$ | $74.48 \pm 0.51$ | $63.90$ |
| DSSL | $44.48 \pm 0.33$ | $40.44 \pm 0.16$ | $\underline{82.05 \pm 0.50}$ | $73.88 \pm 0.76$ | $69.08 \pm 0.82$ | $61.99$ |
| **PROP-DGI** | $74.66 \pm 0.27$ | $\underline{43.14 \pm 0.28}$ | $80.50 \pm 0.62$ | $\underline{77.93 \pm 0.54}$ | $74.88 \pm 0.76$ | $\underline{70.22}$ |

## 8 CONCLUSION

In this work, we suggest a training-free method PROP as a strong baseline in GCL. From the decoupling perspective, we observe that transformation weights learned through GCL present a quite smooth and uninformative characteristic. We further propose to only learn the propagation coefficients in the encoder of GCL, which achieves state-of-the-art performance on diverse no classification benchmarks. We believe that this work opens new avenues for exploring lightweight and effective graph contrastive learning methods, with broad implications for both research and practical applications in the field of graph learning.

## 9 LIMITATIONS

Our study highlights the strong performance of the propagation-only PROP method on diverse node classification benchmarks, showcasing its simplicity and effectiveness. For graph classification tasks, which may involve low-quality or absent node features, PROP offers an initial approach that can be further adapted (see Appendix A). Building on PROP, PROPGCL introduces learnable propagation through spectral filters, making it particularly effective for single-graph tasks. Future research can explore extending PROPGCL to multi-graph settings and enhancing its applicability across diverse graph structures.

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

Table 8: Comparison of training time per epoch in seconds and memory consumption of encoder in KBs between GRACE and PROP-GRACE. *Improvement* refers to the percentage increase in speed or decrease in the memory consumption.

| Metric | Method | Cora | CiteSeer | PubMed | Photo | Computers | CS | Squirrel | Chameleon | Actor |
|--------|--------|------|----------|--------|-------|-----------|-----|----------|-----------|-------|
| | GRACE | 0.1611 | 0.1939 | 0.2795 | 0.2872 | 0.4639 | 1.5111 | 0.7004 | 0.2295 | 0.2872 |
| Time | PROP-GRACE | 0.1409 | 0.1478 | 0.2650 | 0.2400 | 0.3626 | 0.2374 | 0.2581 | 0.1450 | 0.2073 |
| | *Improvement* | 12.54% | 23.79% | 5.18% | 16.44% | 21.84% | 84.29% | 63.15% | 36.82% | 27.83% |
| | GRACE | 3894.04 | 8434.04 | 2028.04 | 2518.04 | 2562.04 | 2562.04 | 5206.04 | 5678.04 | 2892.04 |
| Memory | PROP-GRACE | 11.24 | 28.97 | 3.95 | 5.86 | 6.04 | 6.04 | 16.36 | 18.21 | 7.32 |
| | *Improvement* | 99.71% | 99.66% | 99.81% | 99.77% | 99.76% | 99.76% | 99.69% | 99.68% | 99.75% |

Anthony J Bell and Terrence J Sejnowski. The "independent components" of natural scenes are edge filters. *Vision research*, 37(23):3327–3338, 1997. 7, 19

Ella Bingham and Heikki Mannila. Random projection in dimensionality reduction: applications to image and text data. In *SIGKDD*, 2001. 5

Karsten M Borgwardt, Cheng Soon Ong, Stefan Schönauer, SVN Vishwanathan, Alex J Smola, and Hans-Peter Kriegel. Protein function prediction via graph kernels. *Bioinformatics*, 21(suppl_1): i47–i56, 2005. 15, 25

Jialu Chen and Gang Kou. Attribute and structure preserving graph contrastive learning. In *AAAI*, 2023. 4

Jingfan Chen, Guanghui Zhu, Yifan Qi, Chunfeng Yuan, and Yihua Huang. Towards self-supervised learning on graphs with heterophily. In *CIKM*, 2022. 6, 8, 26, 27

Jingyu Chen, Runlin Lei, and Zhewei Wei. Polygcl: Graph contrastive learning via learnable spectral polynomial filters. In *ICLR*, 2024. 4, 6, 8, 26, 27

Eli Chien, Jianhao Peng, Pan Li, and Olgica Milenkovic. Adaptive universal generalized pagerank graph neural network. In *ICLR*, 2021. 3, 4, 21, 23

Asim Kumar Debnath, Rosa L Lopez de Compadre, Gargi Debnath, Alan J Shusterman, and Corwin Hansch. Structure-activity relationship of mutagenic aromatic and heteroaromatic nitro compounds. correlation with molecular orbital energies and hydrophobicity. *Journal of medicinal chemistry*, 34 (2):786–797, 1991. 15, 25

Michaël Defferrard, Xavier Bresson, and Pierre Vandergheynst. Convolutional neural networks on graphs with fast localized spectral filtering. In *NeurIPS*, 2016. 21

Paul D Dobson and Andrew J Doig. Distinguishing enzyme structures from non-enzymes without alignments. *Journal of molecular biology*, 330(4):771–783, 2003. 15, 25

Hande Dong, Jiawei Chen, Fuli Feng, Xiangnan He, Shuxian Bi, Zhaolin Ding, and Peng Cui. On the equivalence of decoupled graph convolution network and label propagation. In *WWW*, 2021. 3

Yoav Freund, Sanjoy Dasgupta, Mayank Kabra, and Nakul Verma. Learning the structure of manifolds using random projections. In *NeurIPS*, 2007. 5

Yuan Gao, Xin Li, and Yan Hui. Rethinking graph contrastive learning: An efficient single-view approach via instance discrimination. *IEEE Transactions on Multimedia*, 26:3616–3625, 2023. 4

Johannes Gasteiger, Aleksandar Bojchevski, and Stephan Günnemann. Predict then propagate: Graph neural networks meet personalized pagerank. In *ICLR*, 2019a. 3, 5

Johannes Gasteiger, Stefan Weißenberger, and Stephan Günnemann. Diffusion improves graph learning. In *Advances in neural information processing systems*, 2019b. 5

Aditya Grover and Jure Leskovec. node2vec: Scalable feature learning for networks. In *SIGKDD*, 2016. 4, 15, 26, 28

Xiaojun Guo, Yifei Wang, Tianqi Du, and Yisen Wang. Contranorm: A contrastive learning perspective on oversmoothing and beyond. In *ICLR*, 2023a. 7, 19, 20

Xiaojun Guo, Yifei Wang, Zeming Wei, and Yisen Wang. Architecture matters: Uncovering implicit mechanisms in graph contrastive learning. In *NeurIPS*, 2023b. 2

Kaveh Hassani and Amir Hosein Khasahmadi. Contrastive multi-view representation learning on graphs. In *ICML*, 2020. 4, 15, 26, 27, 28

Mingguo He, Zhewei Wei, Hongteng Xu, et al. Bernnet: Learning arbitrary graph spectral filters via bernstein approximation. In *NeurIPS*, 2021. 3, 21, 23

Mingguo He, Zhewei Wei, and Ji-Rong Wen. Convolutional neural networks on graphs with chebyshev approximation, revisited. In *NeurIPS*, 2022. 3, 22, 24

Xiangnan He, Kuan Deng, Xiang Wang, Yan Li, Yongdong Zhang, and Meng Wang. Lightgcn: Simplifying and powering graph convolution network for recommendation. In *SIGIR*, 2020. 3

Weihua Hu, Matthias Fey, Marinka Zitnik, Yuxiao Dong, Hongyu Ren, Bowen Liu, Michele Catasta, and Jure Leskovec. Open graph benchmark: Datasets for machine learning on graphs. *arXiv preprint arXiv:2005.00687*, 2020. 8

Tianyu Hua, Wenxiao Wang, Zihui Xue, Sucheng Ren, Yue Wang, and Hang Zhao. On feature decorrelation in self-supervised learning. In *CVPR*, 2021. 7, 19

Lei Huang, Dawei Yang, Bo Lang, and Jia Deng. Decorrelated batch normalization. In *CVPR*, 2018. 19

Sergey Ioffe. Batch normalization: Accelerating deep network training by reducing internal covariate shift. *arXiv preprint arXiv:1502.03167*, 2015. 20

Agnan Kessy, Alex Lewin, and Korbinian Strimmer. Optimal whitening and decorrelation. *The American Statistician*, 72(4):309–314, 2018. 19, 20

Thomas N Kipf and Max Welling. Variational graph auto-encoders. In *NeurIPS Workshop on Bayesian Deep Learning*, 2016. 4, 26

Thomas N Kipf and Max Welling. Semi-supervised classification with graph convolutional networks. In *ICLR*, 2017. 3

Namkyeong Lee, Junseok Lee, and Chanyoung Park. Augmentation-free self-supervised learning on graphs. In *AAAI*, 2022. 2

Haifeng Li, Jun Cao, Jiawei Zhu, Qinyao Luo, Silu He, and Xuying Wang. Augmentation-free graph contrastive learning of invariant-discriminative representations. *IEEE Transactions on Neural Networks and Learning Systems*, 2023a. 2

Jintang Li, Wangbin Sun, Ruofan Wu, Yuchang Zhu, Liang Chen, and Zibin Zheng. Oversmoothing: A nightmare for graph contrastive learning? *arXiv preprint arXiv:2306.02117*, 2023b. 2

Mingjie Li, Xiaojun Guo, Yifei Wang, Yisen Wang, and Zhouchen Lin. G$^2$cn: Graph gaussian convolution networks with concentrated graph filters. In *ICML*, 2022a. 5

Ping Li, Trevor J Hastie, and Kenneth W Church. Very sparse random projections. In *SIGKDD*, 2006. 5

Sihang Li, Xiang Wang, An Zhang, Yingxin Wu, Xiangnan He, and Tat-Seng Chua. Let invariant rationale discovery inspire graph contrastive learning. In *ICML*, 2022b. 2

Ningyi Liao, Siqiang Luo, Xiang Li, and Jieming Shi. Ld2: Scalable heterophilous graph neural network with decoupled embeddings. In *NeurIPS*, 2024. 5

Meng Liu, Hongyang Gao, and Shuiwang Ji. Towards deeper graph neural networks. In *SIGKDD*, 2020. 3

Nian Liu, Xiao Wang, Deyu Bo, Chuan Shi, and Jian Pei. Revisiting graph contrastive learning from the perspective of graph spectrum. In *NeurIPS*, 2022. 2

Yue Liu, Xihong Yang, Sihang Zhou, Xinwang Liu, Siwei Wang, Ke Liang, Wenxuan Tu, and Liang Li. Simple contrastive graph clustering. *IEEE Transactions on Neural Networks and Learning Systems*, 2023. 2

Sitao Luan, Chenqing Hua, Minkai Xu, Qincheng Lu, Jiaqi Zhu, Xiao-Wen Chang, Jie Fu, Jure Leskovec, and Doina Precup. When do graph neural networks help with node classification: Investigating the homophily principle on node distinguishability. *arXiv preprint arXiv:2304.14274*, 2023. 16

Yao Ma, Xiaorui Liu, Neil Shah, and Jiliang Tang. Is homophily a necessity for graph neural networks? *arXiv preprint arXiv:2106.06134*, 2021. 16

Haitao Mao, Zhikai Chen, Wei Jin, Haoyu Han, Yao Ma, Tong Zhao, Neil Shah, and Jiliang Tang. Demystifying structural disparity in graph neural networks: Can one size fit all? In *NeurIPS*, 2023. 16

Christopher Morris, Nils M. Kriege, Franka Bause, Kristian Kersting, Petra Mutzel, and Marion Neumann. Tudataset: A collection of benchmark datasets for learning with graphs. In *ICML 2020 Workshop on Graph Representation Learning and Beyond*, 2020. 25

Galileo Namata, Ben London, Lise Getoor, Bert Huang, and UMD EDU. Query-driven active surveying for collective classification. In *MLG*, 2012. 4, 24

Annamalai Narayanan, Mahinthan Chandramohan, Rajasekar Venkatesan, Lihui Chen, Yang Liu, and Shantanu Jaiswal. graph2vec: Learning distributed representations of graphs. arxiv 2017. *arXiv preprint arXiv:1707.05005*, 2017. 15, 28

Chaoxi Niu, Guansong Pang, and Ling Chen. Affinity uncertainty-based hard negative mining in graph contrastive learning. *IEEE Transactions on Neural Networks and Learning Systems*, 2024. 2

Hoang Nt and Takanori Maehara. Revisiting graph neural networks: All we have is low-pass filters. *arXiv preprint arXiv:1905.09550*, 2019. 3

Hongbin Pei, Bingzhe Wei, Kevin Chen-Chuan Chang, Yu Lei, and Bo Yang. Geom-gcn: Geometric graph convolutional networks. In *ICLR2020*, 2020. 4, 17, 25

Bryan Perozzi, Rami Al-Rfou, and Steven Skiena. Deepwalk: Online learning of social representations. In *SIGKDD*, 2014. 4, 26

Oleg Platonov, Denis Kuznedelev, Artem Babenko, and Liudmila Prokhorenkova. Characterizing graph datasets for node classification: Homophily-heterophily dichotomy and beyond. In *NeurIPS*, 2023a. 16

Oleg Platonov, Denis Kuznedelev, Michael Diskin, Artem Babenko, and Liudmila Prokhorenkova. A critical look at the evaluation of gnns under heterophily: Are we really making progress? In *ICLR*, 2023b. 8

Ladislav Rampášek, Michael Galkin, Vijay Prakash Dwivedi, Anh Tuan Luu, Guy Wolf, and Dominique Beaini. Recipe for a general, powerful, scalable graph transformer. In *NeurIPS*, 2022. 15

Joshua Robinson, Ching-Yao Chuang, Suvrit Sra, and Stefanie Jegelka. Contrastive learning with hard negative samples. In *ICLR*, 2021. 2

Benedek Rozemberczki, Carl Allen, and Rik Sarkar. Multi-scale attributed node embedding. *Journal of Complex Networks*, 9(2):cnab014, 2021. 4, 25

Guillaume Salha, Romain Hennequin, and Michalis Vazirgiannis. Keep it simple: Graph autoencoders without graph convolutional networks. In *NeurIPS Graph Representation Learning Workshop*, 2019. 2

Prithviraj Sen, Galileo Namata, Mustafa Bilgic, Lise Getoor, Brian Galligher, and Tina Eliassi-Rad. Collective classification in network data. *AI magazine*, 29(3):93–93, 2008. 4, 24

Oleksandr Shchur, Maximilian Mumme, Aleksandar Bojchevski, and Stephan Günnemann. Pitfalls of graph neural network evaluation. *arXiv preprint arXiv:1811.05868*, 2018. 4, 25

Yekini Shehu, Olaniyi S. Iyiola, and Ferdinard U. Ogbuisi. Iterative method with inertial terms for nonexpansive mappings: applications to compressed sensing. *Numerical Algorithms*, 83(4): 1321–1347, 2020. 4

Nino Shervashidze, SVN Vishwanathan, Tobias Petri, Kurt Mehlhorn, and Karsten Borgwardt. Efficient graphlet kernels for large graph comparison. In *AISTATS*, 2009. 15, 28

Nino Shervashidze, Pascal Schweitzer, Erik Jan Van Leeuwen, Kurt Mehlhorn, and Karsten M Borgwardt. Weisfeiler-lehman graph kernels. *Journal of Machine Learning Research*, 12(9), 2011. 15, 28

Fan-Yun Sun, Jordan Hoffmann, Vikas Verma, and Jian Tang. Infograph: Unsupervised and semi-supervised graph-level representation learning via mutual information maximization. In *ICLR*, 2020. 15, 28, 29

Susheel Suresh, Pan Li, Cong Hao, and Jennifer Neville. Adversarial graph augmentation to improve graph contrastive learning. In *NeurIPS*, 2021. 4, 15, 28

Shantanu Thakoor, Corentin Tallec, Mohammad Gheshlaghi Azar, Mehdi Azabou, Eva L Dyer, Remi Munos, Petar Veličković, and Michal Valko. Large-scale representation learning on graphs via bootstrapping. In *ICLR*, 2022. 4, 26, 27

Puja Trivedi, Ekdeep S Lubana, Mark Heimann, Danai Koutra, and Jayaraman Thiagarajan. Analyzing data-centric properties for graph contrastive learning. In *NeurIPS*, 2022. 2

Petar Velickovic, William Fedus, William L Hamilton, Pietro Liò, Yoshua Bengio, and R Devon Hjelm. Deep graph infomax. In *ICLR*, 2019. 1, 2, 4, 26, 27, 28

Nikil Wale, Ian A Watson, and George Karypis. Comparison of descriptor spaces for chemical compound retrieval and classification. *Knowledge and Information Systems*, 14:347–375, 2008. 15, 25

Haonan Wang, Jieyu Zhang, Qi Zhu, Wei Huang, Kenji Kawaguchi, and Xiaokui Xiao. Single-pass contrastive learning can work for both homophilic and heterophilic graph. In *TMLR*, 2023. 8, 26, 27

Felix Wu, Amauri Souza, Tianyi Zhang, Christopher Fifty, Tao Yu, and Kilian Weinberger. Simplifying graph convolutional networks. In *ICML*, 2019. 3

Jun Xia, Lirong Wu, Ge Wang, and Stan Z. Li. Progcl: Rethinking hard negative mining in graph contrastive learning. In *ICML*, 2022. 4, 26, 27

Teng Xiao, Zhengyu Chen, Zhimeng Guo, Zeyang Zhuang, and Suhang Wang. Decoupled self-supervised learning for graphs. In *NeurIPS*, 2022. 8, 26, 27

Teng Xiao, Huaisheng Zhu, Zhengyu Chen, and Suhang Wang. Simple and asymmetric graph contrastive learning without augmentations. In *NeurIPS*, 2024. 8, 26, 27

Dongkuan Xu, Wei Cheng, Dongsheng Luo, Haifeng Chen, and Xiang Zhang. Infogcl: Information-aware graph contrastive learning. In *NeurIPS*, 2021. 2

Pinar Yanardag and SVN Vishwanathan. Deep graph kernels. In *SIGKDD*, 2015. 15, 25, 28

Haoran Yang, Hongxu Chen, Sixiao Zhang, Xiangguo Sun, Qian Li, Xiangyu Zhao, and Guandong Xu. Generating counterfactual hard negative samples for graph contrastive learning. In *WWW*, 2023. 2

Chengxuan Ying, Tianle Cai, Shengjie Luo, Shuxin Zheng, Guolin Ke, Di He, Yanming Shen, and Tie-Yan Liu. Do transformers really perform badly for graph representation? In *NeurIPS*, 2021. 15

Yuning You, Tianlong Chen, Yongduo Sui, Ting Chen, Zhangyang Wang, and Yang Shen. Graph contrastive learning with augmentations. In *NeurIPS*, 2020. 1, 15, 26, 28, 29

Yuning You, Tianlong Chen, Yang Shen, and Zhangyang Wang. Graph contrastive learning automated. In *ICML*, 2021. 15, 28

Junliang Yu, Hongzhi Yin, Xin Xia, Tong Chen, Lizhen Cui, and Quoc Viet Hung Nguyen. Are graph augmentations necessary? simple graph contrastive learning for recommendation. In *SIGIR*, 2022. 2

Yunfeng Yu, Longlong Lin, Qiyu Liu, Zeli Wang, Xi Ou, and Tao Jia. Gsd-gnn: Generalizable and scalable algorithms for decoupled graph neural networks. In *ICMR*, 2024. 5

Seongjun Yun, Minbyul Jeong, Raehyun Kim, Jaewoo Kang, and Hyunwoo J Kim. Graph transformer networks. In *Advances in neural information processing systems*, 2019. 15

Hengrui Zhang, Qitian Wu, Junchi Yan, David Wipf, and Philip S Yu. From canonical correlation analysis to self-supervised graph neural networks. In *NeurIPS*, 2021. 4, 17, 26, 27

Muhan Zhang and Yixin Chen. Link prediction based on graph neural networks. In *NeurIPS*, volume 31, 2018. 1

Yizhen Zheng, Shirui Pan, Vincent Lee, Yu Zheng, and Philip S Yu. Rethinking and scaling up graph contrastive learning: An extremely efficient approach with group discrimination. In *NeurIPS*, 2022. 2, 8

Jiong Zhu, Yujun Yan, Lingxiao Zhao, Mark Heimann, Leman Akoglu, and Danai Koutra. Beyond homophily in graph neural networks: Current limitations and effective designs. In *NeurIPS*, 2020a. 16, 25

Meiqi Zhu, Xiao Wang, Chuan Shi, Houye Ji, and Peng Cui. Interpreting and unifying graph neural networks with an optimization framework. In *WWW*, 2021a. 3

Yanqiao Zhu, Yichen Xu, Feng Yu, Qiang Liu, Shu Wu, and Liang Wang. Deep Graph Contrastive Representation Learning. In *ICML Workshop on Graph Representation Learning and Beyond*, 2020b. 4, 26, 27, 28

Yanqiao Zhu, Yichen Xu, Qiang Liu, and Shu Wu. An empirical study of graph contrastive learning. In *NeurIPS Track on Datasets and Benchmarks.*, 2021b. 2, 26

Yanqiao Zhu, Yichen Xu, Feng Yu, Qiang Liu, Shu Wu, and Liang Wang. Graph contrastive learning with adaptive augmentation. In *WWW*, 2021c. 4, 17, 26, 27

## A EXPERIMENTS OF PROP ON GRAPH CLASSIFICATION

**Methods.** We first aggregate node features within $K$-hop neighbors without any trainable weights, then pool aggregated node features into a global graph representation, *i.e.*,

$$\mathbf{H}_{\text{PROP}} = \frac{1}{N} \sum_i \mathbf{H}_i, \quad \mathbf{H} = \tilde{\mathbf{A}}^K \mathbf{X}, \tag{6}$$

where $N$ is the number of nodes, $\tilde{\mathbf{A}} = \mathbf{D}'^{-\frac{1}{2}} \mathbf{A}' \mathbf{D}'^{-\frac{1}{2}}$ with $\mathbf{A}' = \mathbf{A} + \mathbf{I}$.

**Datasets.** For the graph classification task, we choose molecules datasets MUTAG (Debnath et al., 1991) and NCI1 (Wale et al., 2008), bioinformatics datasets PROTEINS (Borgwardt et al., 2005), and DD (Dobson & Doig, 2003), social networks IMDB-BINARY, IMDB-MULTI (Yanardag & Vishwanathan, 2015), and COLLAB (Yanardag & Vishwanathan, 2015).

**Baselines.** We consider three categories of representative methods as baselines: 1) graph kernel methods including GL (Shervashidze et al., 2009), WL (Shervashidze et al., 2011), and DGK (Yanardag & Vishwanathan, 2015), 2) traditional graph embedding methods including node2vec (Grover & Leskovec, 2016), sub2vec (Adhikari et al., 2018), and graph2vec (Narayanan et al., 2017), 3) contrastive learning methods including InfoGraph (Sun et al., 2020), GraphCL (You et al., 2020), MVGRL (Hassani & Khasahmadi, 2020), JOAOv2 (You et al., 2021), ADGCL (Suresh et al., 2021).

**Settings.** Following (You et al., 2020), we train the model in an unsupervised manner and feed the learned representation into a downstream SVM classifier. To keep comparison fairness, we tune hyperparameters in a unified combination, and keep the search space among methods as consistent as possible. Details can be found in Appendix O.

**Results.** As shown in Table 9, although free of training, PROP surpasses most graph kernels and traditional embeddings, and performs comparably with GCL methods. On average, the mean performance gap between PROP and the best method across datasets is only 2.82%. The results show the potential of PROP on the graph classification task. **Notably, common graph classification benchmarks often have less informative node features than node classification benchmarks, even lacking node attribute description as seen in Table 24.** This probably impedes the ability of PROP. An optional choice is utilizing Laplacian positional embeddings or random-walk embeddings as widely discussed in the literature of graph Transforms (Yun et al., 2019; Ying et al., 2021; Rampášek et al., 2022). We leave deeper research on graph classification tasks for future work.

Table 9: Test accuracy (%) of graph classification benchmarks, comparing PROP and GSSL methods. The compared results are from published papers, and − indicates that results are unavailable. We report the performance gap between one method and the best method, averaged across datasets in the **Mean Gap.** column. Red indicates the best method, while underlined represents the second-best.

| | PROTEINS | MUTAG | DD | NCI1 | IMDB-B | IMDB-M | COLLAB | Mean Gap. ↓ |
|---|---|---|---|---|---|---|---|---|
| *Graph Kernel* | | | | | | | | |
| GL | − | $81.66 \pm 2.11$ | − | − | $65.87 \pm 0.98$ | − | − | 7.60 |
| WL | $72.92 \pm 0.56$ | $80.72 \pm 3.00$ | − | $\underline{80.01 \pm 0.50}$ | $72.30 \pm 3.44$ | − | − | 2.88 |
| DGK | $73.30 \pm 0.82$ | $87.44 \pm 2.72$ | − | $\mathbf{80.31 \pm 0.46}$ | $66.96 \pm 0.56$ | − | − | 2.37 |
| *Traditional Graph Embedding* | | | | | | | | |
| node2vec | $57.49 \pm 3.57$ | $72.63 \pm 10.20$ | − | $54.89 \pm 1.61$ | − | − | − | 16.61 |
| sub2vec | $53.03 \pm 5.55$ | $61.05 \pm 15.80$ | − | $52.84 \pm 1.47$ | $55.26 \pm 1.54$ | − | − | 19.79 |
| graph2vec | $73.30 \pm 2.05$ | $83.15 \pm 9.25$ | − | $73.22 \pm 1.81$ | $71.10 \pm 0.54$ | − | − | 3.54 |
| *Graph Contrastive Learning* | | | | | | | | |
| MVGRL | − | $75.40 \pm 7.80$ | − | − | $63.60 \pm 4.20$ | − | − | 11.87 |
| InfoGraph | $\mathbf{74.44 \pm 0.31}$ | $\underline{89.01 \pm 1.13}$ | $72.85 \pm 1.78$ | $76.20 \pm 1.06$ | $\mathbf{73.03 \pm 0.87}$ | $48.66 \pm 0.67$ | $70.65 \pm 1.13$ | 2.07 |
| GraphCL | $\underline{74.39 \pm 0.45}$ | $86.80 \pm 1.34$ | $\mathbf{78.62 \pm 0.40}$ | $77.87 \pm 0.41$ | $71.14 \pm 0.44$ | $\underline{48.49 \pm 0.63}$ | $\underline{71.36 \pm 1.15}$ | **1.52** |
| JOAOv2 | $74.07 \pm 1.10$ | $87.67 \pm 0.79$ | $77.40 \pm 1.15$ | $78.36 \pm 0.53$ | $70.83 \pm 0.25$ | − | $69.33 \pm 0.34$ | $\underline{1.78}$ |
| ADGCL | $73.81 \pm 0.46$ | $\mathbf{89.70 \pm 1.03}$ | $75.10 \pm 0.39$ | $69.67 \pm 0.51$ | $\underline{72.33 \pm 0.56}$ | $\mathbf{49.89 \pm 0.66}$ | $\mathbf{73.32 \pm 0.61}$ | 2.21 |
| **PROP** | $71.07 \pm 0.30$ | $87.44 \pm 1.53$ | $\underline{78.39 \pm 0.37}$ | $75.24 \pm 0.14$ | $71.22 \pm 0.28$ | $47.11 \pm 0.18$ | $69.07 \pm 0.05$ | 2.82 |

## B  GRAPH STRUCTURE AS SUPERVISED SIGNAL

The taxonomy of homophily and heterophily is widely used to tell whether the graph structure is informative for training GCN-like models. Beyond the discussion on homophily and heterophily, recent metrics characterizing graphs are proposed and show closer relationships with the GNN performance (Mao et al., 2023; Luan et al., 2023; Platonov et al., 2023a). For example, Ma et al. (2021) claim that the inter-class similarity on Squirrel is slightly higher than the intra-class similarity for most classes, which substantiates the middling performance of GCN.

However, the performance of GCN-like models is an interplay between graph structure and node features. Therefore, a bad GCN performance can not indicate the helplessness of graph structure, or vice versa. For verification, we design experiments based on the mutual information of labels and different graph elements. To escape from the entanglement of structure and node features, we use MLP instead of GCN as the trainable model with node features $\mathbf{X}$, adjacency matrix $\mathbf{A}$, and the concatenation of the two as inputs, respectively. The correspondence is as follows:

- $I(\mathbf{Y}; \mathbf{X})$: MLP with $\mathbf{X}$ as inputs.

- $I(\mathbf{Y}; \mathbf{A})$: MLP with $\mathbf{A}$ as inputs.

- $I(\mathbf{Y}; \mathbf{X}; \mathbf{A})$: MLP with $[\mathbf{X}, \mathbf{A}]$ as inputs, where $[]$ denotes concatenation.

The results are shown in Table 10. It is surprising that **for some heterophily datasets, MLP with the graph structure as inputs gets satisfying performance**. For example, for the Squirrel dataset with a low homophily ratio of 0.22, MLP based on the graph structure achieves 73.58% accuracy. Therefore, even presenting a low homophily ratio, the graph structure can still serve as a highly qualified supervision signal for predicting labels.

Table 10: Test accuracy (%) of MLP with different input signals on node classification benchmarks. $\mathcal{H}(G)$ denotes the edge homophily ratio introduced in Zhu et al. (2020a). Lower $\mathcal{H}(G)$ denotes graphs with a high heterophily level. **Bold** indicates the best, while underlined represents the second-best choice.

|  | Cora | CiteSeer | PubMed | Chameleon | Squirrel | Actor |
|---|---|---|---|---|---|---|
| $\mathcal{H}(G)$ | 0.81 | 0.74 | 0.80 | 0.23 | 0.22 | 0.22 |
| MLP($\mathbf{X}$) | 73.64 | 70.72 | 85.75 | 49.34 | 35.06 | **36.51** |
| MLP($\mathbf{A}$) | 78.27 | 57.81 | 81.41 | **77.41** | **73.58** | 21.84 |
| MLP($[\mathbf{X}, \mathbf{A}]$) | **82.29** | **73.57** | **85.83** | 71.05 | 67.63 | 31.84 |

## C  TRIALS IN FEW-SHOT LEARNING

In Section 5, we observe that GCL has the potential to learn good propagation coefficients. It inspires methods in the *few-shot* scenario, where a model is tasked with achieving effective generalization from a minimal number of labeled examples per class.

In this study, we examine the $N$-shot case where $N$ support examples are used for training. As baselines, we evaluate the ChebNetII model trained with both supervised learning (SL) and contrastive learning (CL). As shown in Table 11, SL exhibits low accuracy due to sparse labeling, while CL performs relatively better, given access to all provided samples.

Based on our findings, we first train the ChebNetII model using contrastive learning. We then fix the propagation coefficients learned in GCL and focus on optimizing the transformation weights through a supervised objective. We term the method as ***Fix-prop SL***. As illustrated in Table 11, this approach yields improvements on several benchmarks. For instance, Fix-prop SL enhances SL accuracy from 57.51% to 72.60% on Cora in the 5-shot case, and from 39.19% to 65.39% in the 3-shot case. The results demonstrate the potential of integrating SL and CL from a decoupling perspective in few-shot learning. However, the Fix-prop SL approach has minimal impact on the Squirrel and Chameleon datasets. It is important to note that we keep hyperparameters consistent across all training methods and benchmarks, leaving ample room for further exploration beyond this initial investigation.

Table 11: Test accuracy (%) of node classification benchmarks in the few-shot scenario. **Bold** indicates the best, while underlined represents the second-best choice.

| | Training | Cora | CiteSeer | PubMed | Squirrel | Chameleon |
|---|---|---|---|---|---|---|
| | SL | $57.51 \pm 2.29$ | $43.11 \pm 3.75$ | $59.62 \pm 2.56$ | $20.15 \pm 0.30$ | $22.09 \pm 1.60$ |
| 5 Shot | CL | $66.88 \pm 2.29$ | $\mathbf{55.02 \pm 4.64}$ | $63.20 \pm 2.64$ | $\mathbf{28.41 \pm 0.87}$ | $\mathbf{36.92 \pm 2.52}$ |
| | Fix-prop SL | $\mathbf{72.60 \pm 1.43}$ | $53.26 \pm 4.03$ | $\mathbf{67.66 \pm 2.58}$ | $20.60 \pm 0.90$ | $23.30 \pm 1.91$ |
| | SL | $39.19 \pm 3.96$ | $37.52 \pm 2.25$ | $55.89 \pm 2.55$ | $20.27 \pm 0.55$ | $21.40 \pm 1.26$ |
| 3 Shot | CL | $64.46 \pm 4.34$ | $\mathbf{55.85 \pm 5.15}$ | $59.88 \pm 3.49$ | $\mathbf{25.89 \pm 1.54}$ | $\mathbf{36.12 \pm 1.34}$ |
| | Fix-prop SL | $\mathbf{65.39 \pm 2.15}$ | $46.90 \pm 3.40$ | $\mathbf{61.46 \pm 5.49}$ | $20.38 \pm 0.69$ | $27.85 \pm 3.02$ |

## D  EXTENSIVE EXPERIMENTS OF SECTION 5.1

In Section 5.1, we show that in the GRACE method, after replacing the trained transformation weights with a random Gaussian matrix, the downstream performance does not deteriorate as expected. We conclude that the transformation weights learned in GCL are not better than random.

To enhance the generalizability of our conclusion, we extended our experimental evaluations to include additional GCL methods. The experimental settings are kept the same. Table 12 and Table 13 respectively show the results using the DGI and BGRL methods. For DGI, after replacing the transformation weights $\mathbf{W}_1$ or $\mathbf{W}_2$ with a random Gaussian matrix, the performance is comparable with before. Moreover, replacing both $\mathbf{W}_1$ and $\mathbf{W}_2$ raises the performance from 71.92% to 72.18% on average. For BGRL, substituting the original transformation weights with random matrices brings an increase of nearly 2% in average performance at best. Although we can not exhaustively try all GCL methods, the results of the representative methods are able to verify that GCL fails to learn effective transformation weights.

Table 12: Test accuracy (%) of node classification benchmarks, comparing the transformation weights ($\mathbf{W}_1$ and/or $\mathbf{W}_2$) learned in DGI with random weights. Red indicates the best method, while underlined represents the second-best choice.

| Method | Cora | CiteSeer | PubMed | Squirrel | Chameleon | Texas | Wisconsin | Cornell | **Mean** |
|---|---|---|---|---|---|---|---|---|---|
| DGI | $83.10 \pm 1.10$ | $66.18 \pm 1.30$ | $82.47 \pm 0.38$ | $\color{red}{41.55 \pm 0.78}$ | $61.75 \pm 1.64$ | $85.57 \pm 2.95$ | $74.00 \pm 2.75$ | $80.82 \pm 1.97$ | 71.93 |
| Randomize $\mathbf{W}_1$ | $79.75 \pm 0.80$ | $65.59 \pm 0.60$ | $82.66 \pm 0.39$ | $38.65 \pm 0.87$ | $66.04 \pm 0.85$ | $85.41 \pm 1.97$ | $\color{red}{75.88 \pm 3.75}$ | $80.82 \pm 1.80$ | 71.85 |
| Randomize $\mathbf{W}_2$ | $\color{red}{83.61 \pm 0.92}$ | $\color{red}{70.19 \pm 0.97}$ | $82.56 \pm 0.30$ | $39.38 \pm 1.09$ | $60.20 \pm 1.31$ | $\color{red}{85.74 \pm 3.11}$ | $73.38 \pm 1.63$ | $\color{red}{80.98 \pm 1.97}$ | 72.01 |
| Randomize both | $80.99 \pm 0.77$ | $65.85 \pm 0.60$ | $\color{red}{82.89 \pm 0.37}$ | $41.04 \pm 0.94$ | $\color{red}{68.21 \pm 1.20}$ | $84.92 \pm 3.11$ | $72.75 \pm 1.00$ | $80.82 \pm 1.97$ | $\mathbf{72.18}$ |

Table 13: Test accuracy (%) of node classification benchmarks, comparing the transformation weights ($\mathbf{W}_1$ and/or $\mathbf{W}_2$) learned in BGRL with random weights. Red indicates the best method, while underlined represents the second-best choice.

| Method | Cora | CiteSeer | PubMed | Squirrel | Chameleon | Texas | Wisconsin | Cornell | **Mean** |
|---|---|---|---|---|---|---|---|---|---|
| BGRL | $79.57 \pm 0.90$ | $68.88 \pm 1.36$ | $83.11 \pm 0.40$ | $\color{red}{32.92 \pm 0.39}$ | $46.02 \pm 1.90$ | $\color{red}{85.74 \pm 3.11}$ | $72.75 \pm 2.00$ | $80.49 \pm 1.64$ | 68.69 |
| Randomize $\mathbf{W}_1$ | $81.02 \pm 0.64$ | $\color{red}{71.56 \pm 1.30}$ | $83.11 \pm 0.40$ | $30.48 \pm 0.70$ | $46.26 \pm 1.27$ | $85.25 \pm 1.97$ | $\color{red}{85.63 \pm 3.00}$ | $\color{red}{80.98 \pm 1.97}$ | $\mathbf{70.54}$ |
| Randomize $\mathbf{W}_2$ | $\color{red}{82.97 \pm 1.05}$ | $70.22 \pm 1.02$ | $83.29 \pm 0.38$ | $32.42 \pm 0.79$ | $\color{red}{46.76 \pm 1.29}$ | $85.41 \pm 3.11$ | $72.38 \pm 2.00$ | $80.49 \pm 1.80$ | 69.24 |
| Randomize both | $81.86 \pm 0.61$ | $71.05 \pm 1.06$ | $\color{red}{83.41 \pm 0.41}$ | $30.99 \pm 0.51$ | $46.13 \pm 1.36$ | $85.57 \pm 1.97$ | $72.63 \pm 1.50$ | $\color{red}{80.98 \pm 1.97}$ | 69.08 |

## E  EXPERIMENTS WITH A FIXED PUBLIC-SPLITTING.

In Section 4.2, we evaluate PROP and other graph self-supervised methods on the node classification task with a random splitting. To avoid the conclusion working on one specific split setting, we here evaluate the models on the public fixed splits following Zhu et al. (2021c); Zhang et al. (2021). In practice, we use the public splitting introduced in Pei et al. (2020) for most datasets. There is no available public splitting for Amazon-Photo and Amazon-Computers, so we randomly split the

dataset into 1/1/8 as the train/validation/test set, differing from the splitting in Section 4.2. Other experimental settings are kept the same. As shown in Table 14, on 6 in 10 benchmarks PROP performs the best among baselines and exceeds the runner-up ProGCL by 4.23% on average. The results verify the effectiveness of PROP in different data-splitting cases.

Table 14: Test accuracy (%) of PROP and other graph self-supervised methods on node classification benchmarks with the public splitting. Red indicates the best method, while underlined represents the second-best choice.

| Method | Cora | CiteSeer | PubMed | Photo | Computers | Squirrel | Chameleon | Texas | Wisconsin | Cornell | Mean |
|--------|------|----------|--------|-------|-----------|----------|-----------|-------|-----------|---------|------|
| DeepWalk | 80.87 ± 1.07 | 63.14 ± 1.05 | 81.55 ± 0.27 | 84.66 ± 0.40 | 89.59 ± 0.18 | 43.32 ± 0.79 | 60.81 ± 1.27 | 53.44 ± 5.09 | 43.63 ± 4.25 | 44.59 ± 2.95 | 64.56 |
| Node2Vec | 84.27 ± 0.70 | 66.04 ± 1.83 | 81.33 ± 0.36 | 83.92 ± 0.31 | 89.31 ± 0.20 | 38.41 ± 1.19 | 59.50 ± 2.30 | 60.81 ± 1.89 | 55.10 ± 3.73 | 60.54 ± 3.24 | 67.92 |
| GAE | 85.96 ± 1.03 | 72.78 ± 1.11 | 85.06 ± 0.49 | 75.29 ± 0.53 | 89.50 ± 0.26 | 35.56 ± 1.27 | 56.51 ± 1.62 | 62.43 ± 4.86 | 61.18 ± 3.53 | 60.27 ± 3.51 | 68.45 |
| VGAE | 86.20 ± 0.76 | 73.26 ± 0.65 | 85.19 ± 0.43 | 72.17 ± 0.33 | 86.90 ± 0.38 | 42.38 ± 1.13 | 60.29 ± 1.05 | 63.78 ± 3.51 | 59.61 ± 2.75 | 60.54 ± 2.16 | 69.03 |
| GRACE | 84.10 ± 1.01 | 70.41 ± 0.92 | 84.79 ± 0.38 | 78.51 ± 0.44 | 87.80 ± 0.41 | 39.65 ± 0.87 | 55.83 ± 1.05 | 64.59 ± 4.59 | 58.82 ± 4.91 | 60.81 ± 2.16 | 68.53 |
| DGI | 87.20 ± 0.99 | 72.50 ± 1.49 | 82.55 ± 0.38 | 71.35 ± 0.57 | 80.43 ± 0.63 | 36.61 ± 1.05 | 52.02 ± 1.32 | 70.54 ± 2.97 | 63.53 ± 3.92 | 61.62 ± 2.16 | 67.84 |
| MVGRL | 83.44 ± 0.72 | 71.61 ± 0.73 | 82.48 ± 0.30 | 80.96 ± 0.67 | 86.87 ± 0.41 | 31.48 ± 0.83 | 58.77 ± 1.45 | 68.38 ± 2.98 | 62.94 ± 3.53 | 61.62 ± 2.16 | 68.86 |
| CCA-SSG | 87.71 ± 0.75 | 75.42 ± 0.80 | 85.55 ± 0.40 | 78.96 ± 0.33 | 90.91 ± 0.38 | 40.16 ± 0.74 | 54.98 ± 1.18 | 68.65 ± 3.78 | 64.12 ± 4.31 | 61.89 ± 2.43 | 70.84 |
| BGRL | 85.77 ± 0.89 | 72.66 ± 1.54 | 84.63 ± 0.49 | 74.43 ± 0.91 | 85.50 ± 0.59 | 37.20 ± 1.07 | 53.82 ± 1.67 | 67.03 ± 2.70 | 60.59 ± 3.14 | 60.81 ± 2.43 | 68.24 |
| GCA | 86.60 ± 0.79 | 74.71 ± 1.18 | 86.44 ± 0.34 | 75.63 ± 0.46 | 88.77 ± 0.54 | 41.33 ± 0.88 | 59.28 ± 1.54 | 69.46 ± 2.97 | 62.94 ± 2.75 | 61.89 ± 2.16 | 70.71 |
| ProGCL | 85.45 ± 0.85 | 73.61 ± 1.10 | 86.86 ± 0.41 | 81.64 ± 0.70 | 89.91 ± 0.31 | 50.23 ± 0.86 | 67.81 ± 1.47 | 69.46 ± 2.97 | 62.75 ± 2.75 | 61.35 ± 1.35 | 72.91 |
| PROP | 84.57 ± 0.82 | 74.55 ± 1.09 | 84.65 ± 0.24 | 84.78 ± 0.38 | 90.83 ± 0.34 | 57.20 ± 1.41 | 68.71 ± 1.18 | 71.35 ± 4.60 | 79.61 ± 3.14 | 75.14 ± 3.78 | 77.14 |

## F FLIP EXPERIMENTS IN SECTION 5.2

In this flip experiment, we first train GRACE with ChebNetII as the encoder and save the learned transformation weights $\mathbf{W}_{\mathrm{CL}}$ and propagation coefficients $\boldsymbol{\theta}_{\mathrm{CL}}$. Then we train ChebNetII in the supervised setting with the propagation coefficients fixed with $\boldsymbol{\theta}_{\mathrm{CL}}$, or the transformation weights fixed with $\mathbf{W}_{\mathrm{CL}}$. As shown in Table.15, despite using the propagation coefficients learned by GCL, the model still achieves satisfying performances compared to the original supervised model. However, after replacing the transformation weights, the performance deteriorates largely. The results further confirm our conclusion in Section 5.2

Table 15: Test accuracy (%) of node classification benchmarks. We freeze the propagation coefficients with optimal $\boldsymbol{\theta}_{\mathrm{CL}}$ (or the transformation weights with $\mathbf{W}_{\mathrm{CL}}$), and *learn* the transformation weights (or propagation coefficients) in the supervised setting. $\mathbb{1}$ denotes an all-one vector. Red indicates the best, while underlined represents the second-best choice.

| Method | $\theta$ | $\mathbf{W}$ | Cora | CiteSeer | PubMed | Squirrel | Chameleon | Texas | Wisconsin | Cornell | Mean |
|--------|----------|--------------|------|----------|--------|----------|-----------|-------|-----------|---------|------|
| SL | *Learn* | *Learn* | 88.39 ± 0.74 | 79.67 ± 0.72 | 87.11 ± 0.25 | 49.34 ± 1.09 | 69.52 ± 0.96 | 89.67 ± 2.13 | 91.25 ± 2.75 | 88.36 ± 3.11 | 80.41 |
| CL | $\boldsymbol{\theta}_{\mathrm{CL}}$ | $\mathbf{W}_{\mathrm{CL}}$ | 83.42 ± 0.92 | 74.79 ± 0.57 | 84.92 ± 0.26 | 37.90 ± 0.79 | 55.67 ± 0.96 | 77.87 ± 2.79 | 86.38 ± 3.63 | 75.74 ± 3.61 | 72.09 |
| Fix-transformation | *Learn* | $\mathbf{W}_{\mathrm{CL}}$ | 76.62 ± 2.12 | 76.25 ± 0.64 | 83.32 ± 0.46 | 36.56 ± 0.61 | 52.41 ± 2.06 | 60.16 ± 6.39 | 75.25 ± 4.38 | 59.51 ± 5.08 | 65.01 |
| Fix-propagation | $\boldsymbol{\theta}_{\mathrm{CL}}$ | *Learn* | 87.06 ± 0.53 | 79.55 ± 0.74 | 85.76 ± 0.23 | 41.44 ± 1.06 | 64.44 ± 0.74 | 87.38 ± 2.95 | 90.63 ± 3.00 | 84.26 ± 2.62 | 77.57 |
| All-one baseline | $\mathbb{1}$ | *Learn* | 71.74 ± 3.22 | 75.92 ± 0.61 | 79.38 ± 0.47 | 33.27 ± 0.61 | 42.32 ± 0.90 | 55.41 ± 4.43 | 74.13 ± 4.13 | 60.82 ± 6.56 | 61.65 |

## G AGGREGATION STEP IN PROP

In this section, we present the accuracies of PROP with different propagation steps. We find the best step choice varies among datasets, but a shallow propagation is enough in most cases. As shown in Figure 2, only one-step propagation performs best in datasets including Cora, CiteSeer, Chameleon, Squirrel, Computers, and Photo. For Texas, Wisconsin, Cornell, Actor, and CS, the raw features, (*i.e.*, zero propagation step) are enough. Moreover, when the performance achieves the best, raising the propagation step will cause a degradation.

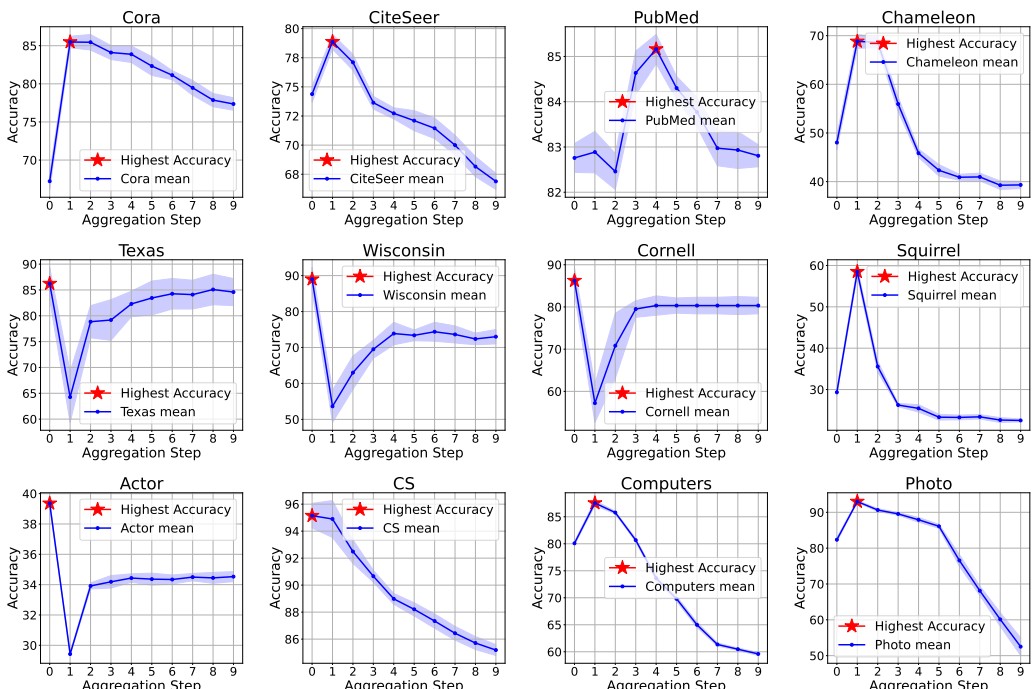

Figure 2: Accuracy (%) of PROP with different propagation steps. We mark the best step choice with a red star. Experiments are conducted ten times and the shadow denotes the derivation.

# H   TRIALS ON LEARNING EFFECTIVE TRANSFORMATION WEIGHTS IN GCL

According to the analysis in Section 5.1, GCL learns uninformative weights that are excessively smoothing. Here we try three ways to solve this problem: 1) enforcing the sparsity of weights with $l_1$ normalization; 2) using whitening methods (Bell & Sejnowski, 1997; Kessy et al., 2018); 3) using normalization methods (Huang et al., 2018; Hua et al., 2021; Guo et al., 2023a).

$l_1$ **regularization.** As a typical technique, the $l_1$ regularization encourages sparsity by driving some weights to zero and retaining the most relevant features. In practice, we add a penalty proportional to the sum of the absolute values of the encoder parameters to the contrastive loss, *i.e.*, $\mathcal{L}_{\text{total}} = \mathcal{L}_{\text{CL}} + \lambda \sum_i |w_i|$, where $\mathcal{L}_{\text{CL}}$ is the contrastive loss, $\lambda$ is the regularization strength, and the $w_i$ is the parameters of the encoder. We conduct experiments on ChebNetII with the $l_1$ regularized GRACE training objective, varying the regularization strength $\lambda$ in $[1 \times 10^{-4}, 1 \times 10^{-5}, 1 \times 10^{-6}]$. As shown in Table 16, the $l_1$ regularization improves performance over the original GRACE on the Squirrel, Chameleon, Texas, Wisconsin, and Cornell datasets, though it still lags behind PROP, except on Wisconsin. However, for Cora, Citeseer, and PubMed, $l_1$ regularization negatively impacts performance.

Table 16: Test accuracy (%) of node classification benchmarks. We train ChebNetII using the $l_1$ regularized GRACE objective. $\lambda$ denotes the regularization strength. Red indicates the best, while underlined represents the second-best choice.

|  | Cora | CiteSeer | PubMed | Squirrel | Chameleon | Texas | Wisconsin | Cornell |
|---|---|---|---|---|---|---|---|---|
| PROP | **85.48 ± 0.76** | **78.87 ± 0.63** | 82.89 ± 0.48 | **58.48 ± 1.03** | **68.82 ± 1.42** | **86.23 ± 3.11** | 89.00 ± 3.25 | **86.23 ± 3.11** |
| λ=0 (GRACE) | 83.42 ± 0.92 | 74.79 ± 0.57 | **84.92 ± 0.26** | 37.90 ± 0.79 | 55.67 ± 0.96 | 77.87 ± 2.79 | 86.38 ± 3.63 | 75.74 ± 3.61 |
| λ=1e-4 | 53.71 ± 1.10 | 26.97 ± 0.50 | 81.20 ± 0.21 | 33.07 ± 0.89 | 48.60 ± 1.42 | 80.98 ± 2.30 | 70.00 ± 1.88 | 82.79 ± 2.46 |
| λ=1e-5 | 78.87 ± 1.17 | 73.29 ± 0.63 | 84.17 ± 0.23 | 37.46 ± 0.89 | 56.37 ± 1.01 | 56.56 ± 1.97 | **91.88 ± 2.25** | 81.80 ± 2.30 |
| λ=1e-6 | 77.75 ± 0.80 | 73.90 ± 0.74 | 84.16 ± 0.21 | 38.27 ± 1.02 | 56.91 ± 1.09 | 52.79 ± 4.76 | 86.88 ± 2.88 | 74.26 ± 7.38 |

**Whitening methods.** Whitening methods are used to decorrelate and normalize data. By making dimensions mutually independent, whitening methods implicitly solve the representation collapse problem. Here we consider the typical Zero-phase Component Analysis (ZCA) whitening (Kessy et al., 2018), which transforms the input data such that it has zero mean and identity covariance matrix, while also preserving data structure as much as possible. It is computed by multiplying the data by the inverse square root of its covariance matrix, *i.e.*, $\hat{x} = \mathbf{V}\mathbf{\Lambda}^{-\frac{1}{2}}\mathbf{V}^{\top}x$, where $\mathbf{V}$ is the matrix of eigenvectors and $\mathbf{\Lambda}$ is the diagonal matrix of eigenvalues of the covariance matrix of $x$. We conduct experiments under the GRACE framework with a ZCA whitening layer added to the encoder ChebNetII. As shown in Table 17, the whitening improves performance over the original GRACE on the PubMed and Chameleon datasets but drastically deteriorates most of the other datasets.

Table 17: Test accuracy (%) of node classification benchmarks. We train ChebNetII using GRACE with the ZCA whitening. Red indicates the best, while underlined represents the second-best choice.

| | Cora | CiteSeer | PubMed | Squirrel | Chameleon | Texas | Wisconsin | Cornell |
|---|---|---|---|---|---|---|---|---|
| PROP | **85.48 ± 0.76** | **78.87 ± 0.63** | 82.89 ± 0.48 | **58.48 ± 1.03** | **68.82 ± 1.42** | **86.23 ± 3.11** | 89.00 ± 3.25 | **86.23 ± 3.11** |
| GRACE | 83.42 ± 0.92 | 74.79 ± 0.57 | 84.92 ± 0.26 | 37.90 ± 0.79 | 55.67 ± 0.96 | 77.87 ± 2.79 | 86.38 ± 3.63 | 75.74 ± 3.61 |
| GRACE+ZCA | 79.29 ± 1.71 | 47.29 ± 0.70 | **85.76 ± 0.29** | 36.72 ± 0.91 | 58.60 ± 1.07 | 43.77 ± 8.36 | 27.38 ± 3.63 | 38.52 ± 6.23 |

**Normalization methods.** For normalization methods, we consider the widely used Batch Normalization (BN) (Ioffe, 2015), and the recently proposed Decorrelate ContraNorm (DCN) (Guo et al., 2023a). Batch normalization scales and shifts the mini-batch of data to have a mean of zero and a standard deviation of one, *i.e.*, $\hat{x} = (x - \mu_B)/\sqrt{\sigma_B^2 + \epsilon}$, where $\mu_B$ and $\sigma_B^2$ are the mean and variance of the mini-batch $B$, and $\epsilon$ is a small constant for numerical stability. DCN scatters representations in the embedding space and leads to a more uniform distribution. The formulation of GCN is $\hat{x} = x - s \times x \times \text{softmax}(x^{\top}x)$, where $s$ is the scale factor. We conduct experiments under the GRACE framework with a BN or DCN layer added to the encoder ChebNetII. As shown in Table 18, BN and DCN both fail to bring substantial improvement over the original GRACE.

Table 18: Test accuracy (%) of node classification benchmarks. We train ChebNetII using GRACE with BN or DCN normalization. $s$ denotes the scale factor in DCN. Red indicates the best, while underlined represents the second-best choice.

| | Cora | CiteSeer | PubMed | Squirrel | Chameleon | Texas | Wisconsin | Cornell |
|---|---|---|---|---|---|---|---|---|
| PROP | **85.48 ± 0.76** | **78.87 ± 0.63** | 82.89 ± 0.48 | **58.48 ± 1.03** | **68.82 ± 1.42** | **86.23 ± 3.11** | 89.00 ± 3.25 | **86.23 ± 3.11** |
| GRACE | 83.42 ± 0.92 | 74.79 ± 0.57 | 84.92 ± 0.26 | 37.90 ± 0.79 | 55.67 ± 0.96 | 77.87 ± 2.79 | 86.38 ± 3.63 | 75.74 ± 3.61 |
| GRACE + BN | 82.25 ± 1.00 | 72.78 ± 1.00 | **85.10 ± 0.24** | 39.56 ± 0.47 | 54.77 ± 0.74 | 76.07 ± 2.95 | 72.63 ± 4.75 | 75.90 ± 2.79 |
| GRACE + DCN ($s$=0.5) | 79.79 ± 0.99 | 73.86 ± 0.86 | 84.00 ± 0.37 | 38.17 ± 0.95 | 56.19 ± 1.03 | 71.15 ± 2.13 | 83.25 ± 2.50 | 71.64 ± 4.59 |
| GRACE + DCN ($s$=1.0) | 75.19 ± 1.08 | 74.91 ± 0.63 | 83.06 ± 0.22 | 38.28 ± 1.12 | 57.35 ± 0.98 | 74.26 ± 1.64 | **90.50 ± 1.50** | 76.72 ± 3.11 |
| GRACE + DCN ($s$=5.0) | 74.40 ± 1.15 | 74.46 ± 0.63 | 79.41 ± 0.35 | 38.01 ± 0.79 | 58.97 ± 1.33 | 72.95 ± 3.44 | 83.25 ± 2.75 | 73.44 ± 3.44 |

In summary, these techniques offer limited effectiveness for GCL when used with polynomial GNNs. We think the possible reason is that the learning of transformation weights needs a high-quality supervision signal. Although these methods help prevent representation collapse, they do not carry extra information. Therefore, GCL still fails to learn good transformation weights.

# I  HYPERPARAMETER SENSITIVITY ANALYSIS

In this section, we conduct the hyperparameter sensitivity analysis comparing PROPGCL and the corresponding backbone GCL methods. We vary the range of hyperparameters and evaluate the downstream performance. Here, we choose two hyperparameters in the model architecture, the hidden dimension and the propagation step. We consider the DGI backbone with the Chebyshev basis. As shown in Figure 3 and Figure 4, the performance of DGI with ChebNetII is highly influenced by disturbing hyperparameters. For example, on Cora, decreasing the hidden dimension from 256 to 128 causes nearly 40% accuracy degradation. In comparison, the performances of PROP-DGI show low variance under different hyperparameter combinations, and a sharp decline is only observed when using small neural networks.

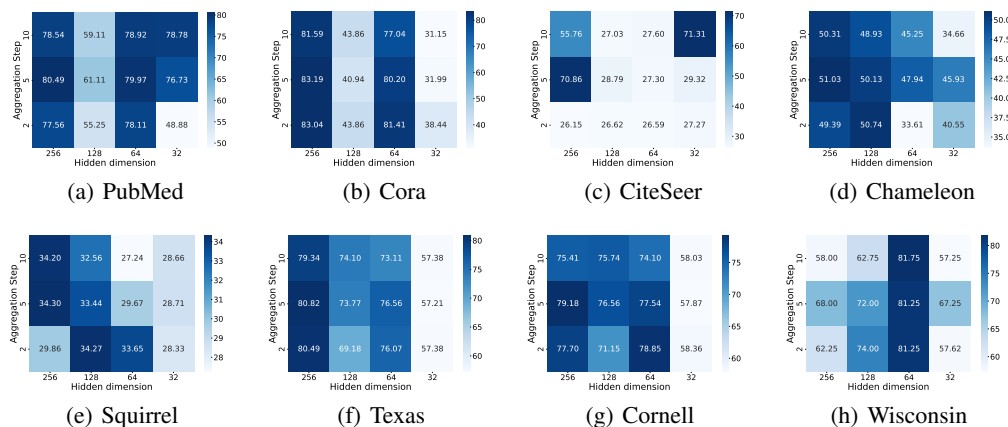

Figure 3: Hyperparameter sensitivity analysis on the hidden dimension and propagation step. Experiments are conducted on DGI with ChebNetII as the encoder.

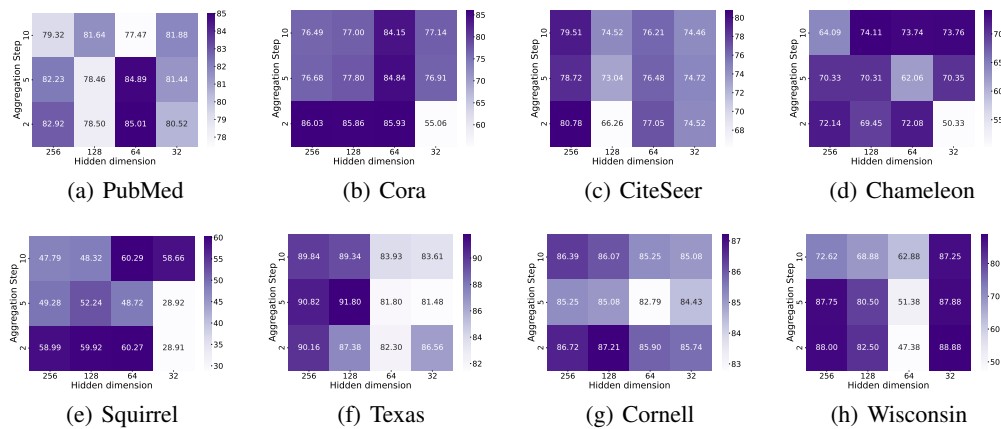

Figure 4: Hyperparameter sensitivity analysis on the hidden dimension and propagation step. Experiments are conducted on PROP-DGI with the Chebyshev basis.

## J  DETAILS ABOUT POLYNOMIAL GNNS

In this section, we introduce polynomial GNNs from the spectral perspective. Developed from graph signal processing, *graph convolution* means transforming the graph signals to the Fourier domain and then back to the vertex domain after suitable filtering, *i.e.*, $\mathbf{H} = \mathbf{U}g_\theta(\mathbf{\Lambda})\mathbf{U}^\top\mathbf{X}$, where $g_\theta$ is the filter, $\mathbf{U}$ is the matrix of eigenvectors of graph Laplacian $\mathbf{L}$, $\mathbf{\Lambda}$ is the diagonal matrix of eigenvalues. The problem arises when the parameters in $g_\theta(\mathbf{\Lambda})$ are entirely unconstrained, leading to a lack of spatial localization in the convolution and a high time complexity due to eigenvalue decomposition.

These issues can be overcome with the use of a polynomial filter $g_\theta(\mathbf{\Lambda}) = \sum_{k=0}^{K-1} \theta_k \mathbf{\Lambda}^k$, where the parameter $\theta \in \mathbb{R}^K$ is a vector of polynomial coefficients. Therefore, the graph convolution can be reformulated as $\mathbf{H} = (\sum_{k=0}^{K-1} \theta_k \mathbf{L}^k)\mathbf{X}$. We call GNNs using the polynomial approximated filters as *polynomial GNNs*. As one of the pioneer works, ChebNet (Defferrard et al., 2016) use Chebyshev polynomial parametrization to localize filters as $g_\theta(\mathbf{\Lambda}) = \sum_{k=0}^{K} \theta_k T_k(\tilde{\mathbf{\Lambda}})$, where $\tilde{\mathbf{\Lambda}} = 2\mathbf{\Lambda}/\lambda_{\max} - \mathbf{I}$, $\theta$ is the Chebyshev coefficients, and $T_k(\tilde{\mathbf{\Lambda}})$ is the Chebyshev polynomial of order $k$ recursively calculated by $T_k(x) = 2xT_{k-1}(x) - T_{k-2}(x)$ with $T_0(x) = 1$ and $T_1(x) = x$.

In section **??**, we consider three popular polynomial GNN variants. GPRGNN (Chien et al., 2021) uses the monomial basis functions evaluated at $\hat{\mathbf{A}}$, *i.e.*, $g_\theta(\mathbf{\Lambda}) = \sum_{k=0}^{K-1} \theta_k(\mathbf{I} - \hat{\mathbf{L}})^k$ with $\theta$ as learnable coefficients. BernNet (He et al., 2021) uses the Bernstein polynomial approximation,

*i.e.*, $g_\theta(\mathbf{\Lambda}) = \sum_{k=0}^{K-1} \theta_k \frac{1}{2^k} \binom{K}{k} (2\mathbf{I} - \mathbf{L})^{K-k} \mathbf{L}^k$ with $\boldsymbol{\theta}$ as learnable coefficients. ChebNetII (He et al., 2022) enhances the original Chebyshev polynomial approximation by Chebyshev interpolation, formulated as $g_\theta(\mathbf{\Lambda}) = \frac{2}{K+1} \sum_{k=0}^{K} \sum_{j=0}^{K} \theta_j T_k(x_j) T_k(\hat{\mathbf{L}})$, where $x_j = \cos((j+1/2)\pi/(K+1))$ are the Chebyshev nodes of $T_{K+1}$, and $\boldsymbol{\theta}$ are learnable coefficients.

## K    CHARACTERIZATION OF LEARNED PROPAGATION COEFFICIENTS

In section 5.2, we find after replacing the transformation weights with supervised ones, the model trained in GCL performs as well as in a supervised manner. To show that given the transformation weights, GCL can learn effective propagation coefficients. We compare the propagation coefficients learned by SL, GCL, and the fix-transformation GCL. As shown in Figure 5, compared with CL, the propagation coefficients learned by fix-transformation GCL are closer to those in SL. Notably, the best propagation coefficients for one dataset may not be unique. Therefore, differing from the SL coefficients does not necessarily indicate poor quality, and the results can not prove that GCL learns bad propagation coefficients. However, it demonstrates that GCL can learn effective propagation coefficients fitting the given transformation weights.

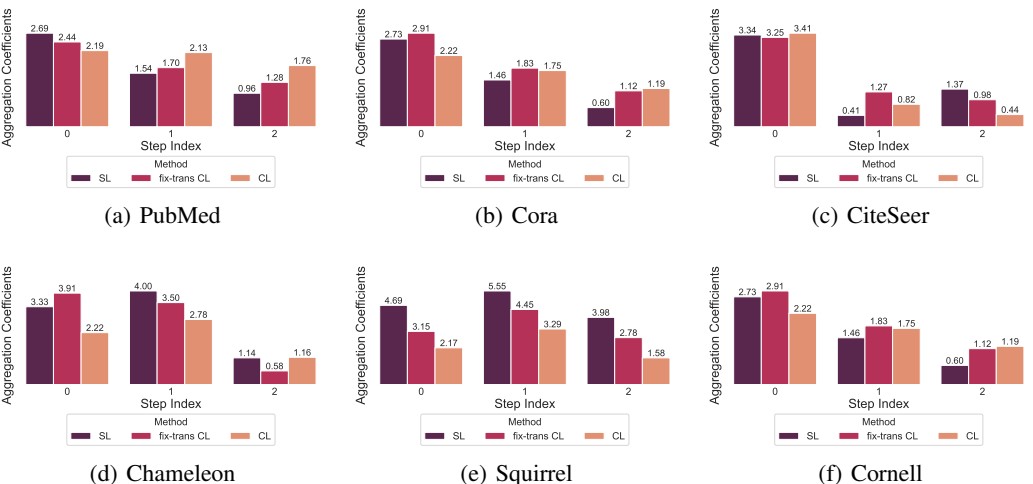

Figure 5: Propagation coefficients of the supervised learning (SL), the contrastive learning (CL), and the fix-transformation contrastive learning (fix-trans CL) introduced in Section 5.2. We show the first three propagation coefficients for the space limit.

## L    CHARACTERIZATION OF LEARNED TRANSFORMATION WEIGHTS

In Section 5.1, we demonstrated the transformation weights learned by GCL and SL on the Cora dataset. Here, we extend these findings by presenting comprehensive results across various datasets. As depicted in Figure 6, the weights learned by GCL exhibit a smoother heatmap compared to those learned by SL. Furthermore, as shown in Figure 7, the weights learned by SL display diverse, data-dependent distributions, while those learned by CL consistently follow a Gaussian-like distribution. These results provide further evidence that GCL struggles to learn effective transformation weights.

## M    EFFICIENCY ANALYSIS

PROPGCL is more efficient than the original baselines in time and memory consumption as shown in Table 19 and Table 20. Remarkably, PRO-GRACE saves 84.29% training time per epoch for the original GRACE with Chebyshev basis on Coauthor CS. For memory consumption, PROP-GRACE consumes over 99% less memory in the encoder for different benchmarks than the original baseline. The boost of time and memory efficiency of PROPGCL is attributed to the exclusion of transformation weights computation in self-supervised training.

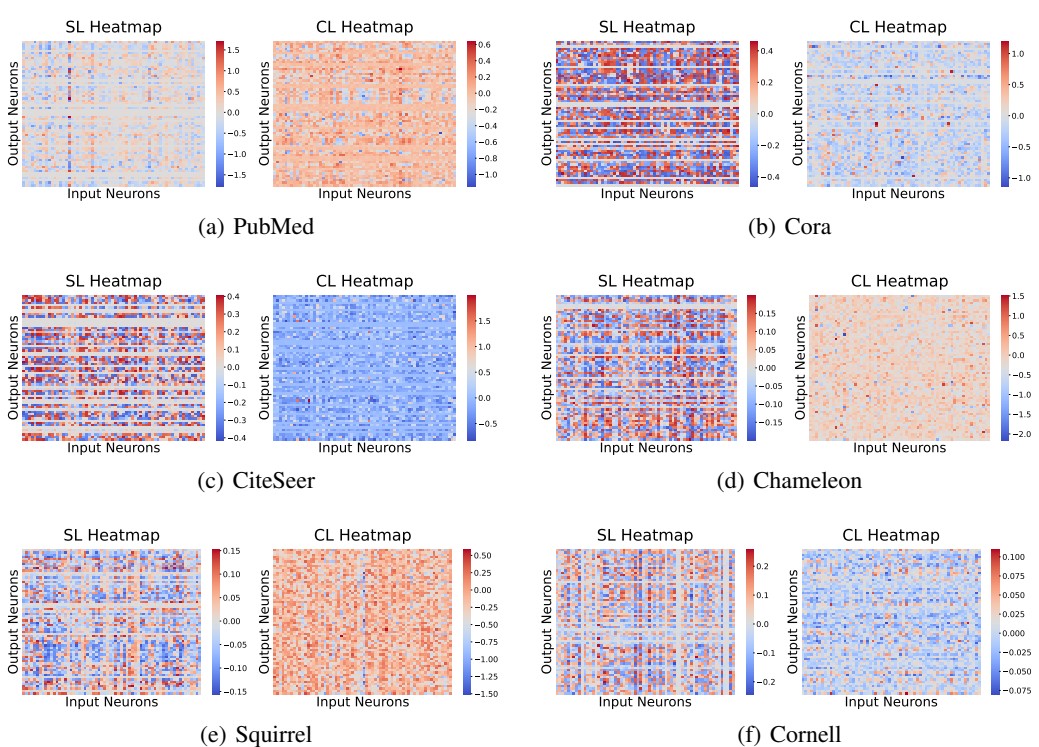

Figure 6: Heatmap of the transformation weights learned by GCL and SL.

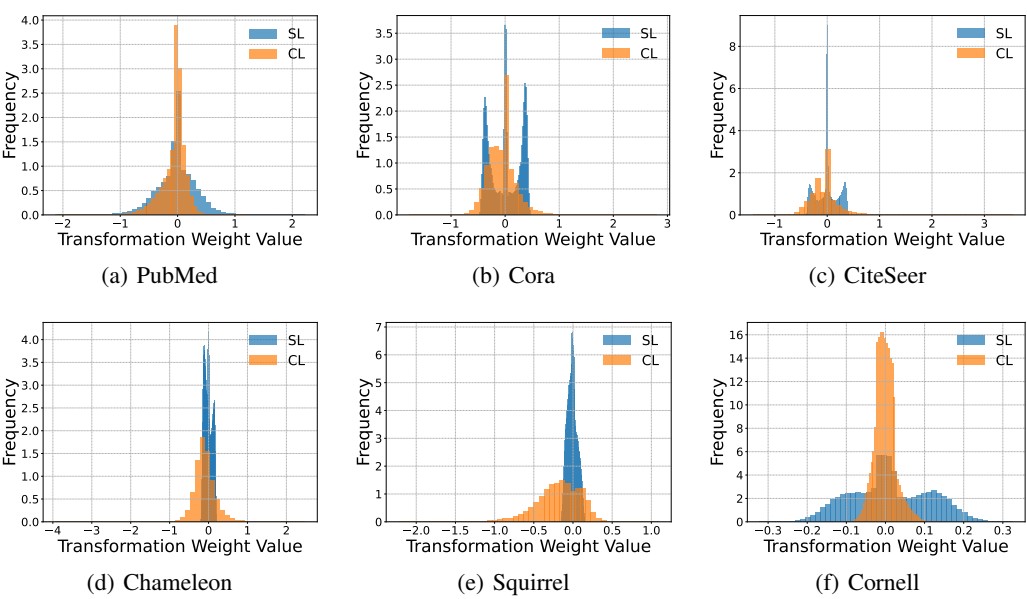

Figure 7: Distribution of the transformation weights learned by GCL and SL.

# N ANALYSIS ON BASIS POLYNOMIAL FUNCTIONS

Polynomial GNNs variants mainly differ in the polynomial basis function choices, *e.g.*, the monomial basis in GPRGNN (Chien et al., 2021), the Bernstein basis in BernNet (He et al., 2021), and the

Table 19: Comparison of training time per epoch in seconds between polynomial GNNs and its corresponding -PROP version in the GRACE framework. *Improvement* refers to the percentage increase in speed of the -PROP version compared to the baseline, *i.e.*, $(t_{\text{GRACE}} - t_{\text{PROP-GRACE}})/t_{\text{GRACE}}$. Experiments are all conducted on a single 24GB NVIDIA GeForce RTX 3090, except those denoted with $*$ on 48GB Nvidia A40 for out-of-memory.

| Basis | Method | Cora | CiteSeer | PubMed | Photo | Computers | CS | Squirrel | Chameleon | Actor |
|---|---|---|---|---|---|---|---|---|---|---|
| Chebyshev | GRACE | 0.1611 | 0.1939 | 0.2795 | 0.2872 | 0.4639 | 1.5111* | 0.7004 | 0.2295 | 0.2872 |
| | PROP-GRACE | 0.1409 | 0.1478 | 0.2650 | 0.2400 | 0.3626 | 0.2374* | 0.2581 | 0.1450 | 0.2073 |
| | *Improvement* | 12.54% | 23.79% | 5.18% | 16.44% | 21.84% | 84.29% | 63.15% | 36.82% | 27.83% |
| Bernstein | GRACE | 0.1515 | 0.2215 | 0.2513 | 0.4878 | 0.9293 | 6.7666* | 1.8997 | 0.4079 | 0.2619 |
| | PROP-GRACE | 0.1226 | 0.1178 | 0.2334 | 0.3832 | 0.6968 | 0.6038* | 0.5175 | 0.1653 | 0.1789 |
| | *Improvement* | 19.03% | 46.79% | 7.10% | 21.45% | 25.02% | 91.08% | 72.76% | 59.47% | 31.69% |
| Monomial | GRACE | 0.1114 | 0.1023 | 0.1217 | 0.1606 | 0.2340 | 1.2487* | 0.3714 | 0.1524 | 0.1202 |
| | PROP-GRACE | 0.1024 | 0.1224 | 0.1221 | 0.1428 | 0.1928 | 0.1927* | 0.1650 | 0.1151 | 0.1109 |
| | *Improvement* | 8.06% | 16.42% | 0.31% | 11.12% | 17.61% | 84.57% | 55.56% | 24.46% | 7.74% |

Table 20: Comparison of memory consumption of encoder in KBs between PROPGCL and the original baseline. We consider GRACE with the Chebyshev basis function here. *Improvement.* refers to the percentage decrease in the memory consumption of the -PROP version compared to the baseline. *i.e.*, $(m_{\text{GRACE}} - m_{\text{PROP-GRACE}})/m_{\text{GRACE}}$.

| Encoder | Cora | CiteSeer | PubMed | Photo | Computers | CS | Squirrel | Chameleon | Actor |
|---|---|---|---|---|---|---|---|---|---|
| GRACE | 3894.04 | 8434.04 | 2028.04 | 2518.04 | 2562.04 | 2562.04 | 5206.04 | 5678.04 | 2892.04 |
| PROP-GRACE | 11.24 | 28.97 | 3.95 | 5.86 | 6.04 | 6.04 | 16.36 | 18.21 | 7.32 |
| *Improvement* | 99.71% | 99.66% | 99.81% | 99.77% | 99.76% | 99.76% | 99.69% | 99.68% | 99.75% |

Chebyshev basis in ChebNetII (He et al., 2022). We introduce detailed basis function formulations in Appendix J.

In this section, we compare different basis polynomial functions used in PROPGCL. Here we consider the Chebyshev basis, Bernstein basis, and monomial basis. As shown in Table 21 and Table 22, the performance of PROPGCL is relatively robust in the choice of basis functions. For homophily benchmarks, PROP-GRACE with Chebyshev basis and the PROP-DGI with monomial basis achieve the best, surpassing the second slightly by 0.05% on average. For heterophily benchmarks, the best PROP-DGI with the Chebyshev basis achieves 73.71% on average, and the Bernstein basis ranks second. In general, the Chebyshev basis is preferred in PROPGCL.

Table 21: Test accuracy (%) of homophily node classification benchmarks, comparing different basis functions in PROPGCL. Red indicates the best method, while underlined represents the second-best.

| Method | Basis | Cora | CiteSeer | PubMed | Photo | Computers | CS | Mean |
|---|---|---|---|---|---|---|---|---|
| PROP-GRACE | Chebyshev | 87.42 ± 0.95 | 81.56 ± 0.83 | 86.19 ± 0.35 | 93.32 ± 0.31 | 88.12 ± 0.23 | 95.95 ± 0.14 | **88.76** |
| | Bernstein | 87.52 ± 1.20 | 81.69 ± 0.86 | 85.90 ± 0.25 | 93.42 ± 0.24 | 87.77 ± 0.22 | 95.97 ± 0.13 | 88.71 |
| | monomial | 87.34 ± 1.13 | 81.86 ± 0.79 | 86.41 ± 0.23 | 93.19 ± 0.26 | 86.85 ± 0.34 | 95.91 ± 0.15 | 88.59 |
| PROP-DGI | Chebyshev | 86.19 ± 1.05 | 80.78 ± 0.65 | 85.14 ± 0.22 | 92.78 ± 0.37 | 89.81 ± 0.20 | 95.82 ± 0.18 | 88.42 |
| | Bernstein | 86.49 ± 0.99 | 80.93 ± 0.72 | 85.80 ± 0.40 | 93.53 ± 0.26 | 89.77 ± 0.25 | 95.46 ± 0.16 | 88.66 |
| | monomial | 86.86 ± 1.02 | 81.69 ± 0.86 | 86.56 ± 0.33 | 93.72 ± 0.25 | 88.18 ± 0.34 | 95.57 ± 0.14 | **88.76** |

## O EXPERIMENTAL DETAILS

### O.1 BENCHMARKS

**Node classification benchmarks**. 1) *Citation Networks* (Sen et al., 2008; Namata et al., 2012). Cora, CiteSeer, and PubMed are three popular citation graph datasets. In these graphs, nodes represent

Table 22: Test accuracy (%) of heterophily node classification benchmarks, comparing different basis functions in PROPGCL. Red indicates the best method, while underlined represents the second-best.

| Method | Basis | Squirrel | Chameleon | Actor | Texas | Wisconsin | Cornell | Mean |
|---|---|---|---|---|---|---|---|---|
| PROP-GRACE | Chebyshev | $55.09 \pm 0.81$ | $71.73 \pm 1.18$ | $39.35 \pm 0.81$ | $89.84 \pm 1.81$ | $88.50 \pm 3.63$ | $86.72 \pm 2.46$ | 71.87 |
| | Bernstein | $48.51 \pm 0.85$ | $70.02 \pm 0.88$ | $39.33 \pm 0.81$ | $90.16 \pm 1.31$ | $\underline{89.00 \pm 3.25}$ | $88.52 \pm 2.95$ | 70.92 |
| | monomial | $51.96 \pm 0.69$ | $69.28 \pm 1.05$ | $\underline{39.52 \pm 0.89}$ | $84.43 \pm 2.62$ | $84.13 \pm 4.50$ | $88.20 \pm 2.79$ | 69.59 |
| PROP-DGI | Chebyshev | $60.53 \pm 0.66$ | $74.11 \pm 0.96$ | $39.53 \pm 0.84$ | $91.80 \pm 2.30$ | $88.88 \pm 2.50$ | $87.38 \pm 2.62$ | 73.71 |
| | Bernstein | $53.08 \pm 0.83$ | $71.20 \pm 0.81$ | $39.48 \pm 0.77$ | $\underline{92.46 \pm 1.48}$ | $91.63 \pm 3.00$ | $87.38 \pm 2.63$ | $\underline{72.54}$ |
| | monomial | $\underline{56.65 \pm 0.77}$ | $\underline{72.12 \pm 0.72}$ | $37.80 \pm 0.57$ | $93.11 \pm 1.80$ | $83.63 \pm 5.88$ | $81.97 \pm 2.95$ | 70.88 |

papers and edges correspond to the citation relationship between two papers. Nodes are classified according to academic topics. 2) *Amazon Co-purchase Networks* (Shchur et al., 2018). Photo and Computers are collected by crawling Amazon websites. Goods are represented as nodes and the co-purchase relationships are denoted as edges. Node features are the bag-of-words representation of product reviews. Each node is labeled with the category of goods. 3) *Wikipedia Networks* (Rozemberczki et al., 2021). Squirrel and Chameleon are collected from the English Wikipedia, representing page-page networks on specific topics. Nodes represent articles and edges are mutual links between them. 4) *WebKB Networks* (Pei et al., 2020). In Texas, Wisconsin, and Cornell datasets, nodes represent web pages and edges represent hyperlinks between them. Node features are the bag-of-words representation of web pages. 5) *Actor Networks* Pei et al. (2020). Each node corresponds to an actor, and the edge between two nodes denotes co-occurrence on the same Wikipedia page. Node features correspond to some keywords on the Wikipedia pages. Statistics of datasets are shown in Table 23.

Table 23: Statistics of node classification benchmarks. $\mathcal{H}(G)$ denotes the edge homophily ratio introduced in Zhu et al. (2020a).

| Homo / Hetero | Category | Dataset | # Nodes | # Edges | # Features | # Classes | $\mathcal{H}(G)$ |
|---|---|---|---|---|---|---|---|
| Homophily | Citation | Cora | 2,708 | 5,278 | 1,433 | 7 | 0.81 |
| | | CiteSeer | 3,327 | 4,552 | 3,703 | 6 | 0.74 |
| | | PubMed | 19,717 | 44,338 | 500 | 3 | 0.80 |
| | Co-purchase | Photo | 7,650 | 119,081 | 745 | 8 | 0.83 |
| | | Computers | 13,752 | 245,861 | 767 | 10 | 0.78 |
| Heterophily | Wikipedia | Chameleon | 2,277 | 36,101 | 2,325 | 6 | 0.23 |
| | | Squirrel | 5,201 | 217,073 | 2,089 | 4 | 0.22 |
| | WebKB | Texas | 183 | 279 | 1703 | 5 | 0.11 |
| | | Wisconsin | 251 | 466 | 1703 | 5 | 0.21 |
| | | Cornell | 183 | 277 | 1703 | 5 | 0.30 |
| | Film-actor | Actor | 7,600 | 30,019 | 932 | 5 | 0.22 |

**Graph Classification benchmarks**. 1) *Molecules*. MUTAG (Debnath et al., 1991) is a dataset of nitroaromatic compounds and the goal is to predict their mutagenicity on Salmonella Typhimurium. NCI1 (Wale et al., 2008) is a dataset of chemical molecules that are annotated based on their activity against non-small cell lung cancer and ovarian cancer cell lines. 2) *Bioinformatics*. PROTEINS (Borgwardt et al., 2005) is a dataset of proteins that are classified as enzymes or non-enzymes. Nodes represent the amino acids and two nodes are connected by an edge if they are less than 6 Angstroms apart. DD (Dobson & Doig, 2003) consists of protein structures with nodes corresponding to amino acids and edges indicating that two amino acids are within a certain number of angstroms. 3) *Social Networks*. IMDB-BINARY and IMDB-MULTI (Yanardag & Vishwanathan, 2015) are movie collaboration datasets consisting of a network of 1,000 actors/actresses who played roles in movies in IMDB. In each graph, nodes represent actors/actresses; corresponding nodes are connected if they appear in the same movie. COLLAB (Yanardag & Vishwanathan, 2015) is derived from three public collaboration datasets representing scientific collaborations between authors. For all benchmarks, we use collections from TUDataset (Morris et al., 2020). Statistics of datasets are shown in Table 24.

Table 24: Statistics of graph classification benchmarks. We report average numbers of nodes, edges, and features across graphs in graph classification datasets.

| Category | Dataset | #Graphs | # Nodes | # Edges | # Features | # Classes |
|---|---|---|---|---|---|---|
| Moleculars | MUTAG | 188 | 17.9 | 39.6 | 7 | 2 |
| | NCI1 | 4110 | 29.87 | 32.30 | 37 | 2 |
| Proteins | PROTEINS | 1113 | 39.1 | 145.6 | 0 | 2 |
| | DD | 1178 | 284.32 | 715.66 | 89 | 2 |
| Social Networks | IMDB-BINARY | 1000 | 19.8 | 193.1 | 0 | 2 |
| | IMDB-MULTI | 1500 | 13.0 | 131.9 | 0 | 3 |
| | COLLAB | 5000 | 74.49 | 2457.78 | 0 | 3 |

## O.2 BASELINES

We categorize baselines for the **node classification task** into 1) traditional graph embedding algorithms DeepWalk (Perozzi et al., 2014) and Node2Vec (Grover & Leskovec, 2016); 2) graph autoencoders GAE (Kipf & Welling, 2016), VGAE (Kipf & Welling, 2016); 3) graph contrastive methods GRACE (Zhu et al., 2020b), DGI (Velickovic et al., 2019), GCA (Zhu et al., 2021c), MV-GRL (Hassani & Khasahmadi, 2020), ProGCL (Xia et al., 2022); 4) graph non-contrastive methods CCA-SSG (Zhang et al., 2021) and BGRL (Thakoor et al., 2022), 5) heterophily baselines compared in Section 6.2, PolyGCL (Chen et al., 2024), HGRL (Chen et al., 2022), GraphACL (Xiao et al., 2024), SP-GCL (Wang et al., 2023), DSSL (Xiao et al., 2022).The design details are as follows.

1) *Traditional graph embeddings*.

- **DeepWalk** (Perozzi et al., 2014). DeepWalk leverages truncated random walks to capture local network structures. The algorithm treats the random walks as sequences of nodes, akin to sentences in language models. It learns latent representations by applying skip-gram to maximize the co-occurrence probabilities of nodes appearing in these random walks.
- **Node2Vec** (Grover & Leskovec, 2016). Node2Vec is built on DeepWalk by introducing a flexible biased random walk strategy to explore network neighborhoods. The key innovation is balancing breadth-first sampling (BFS) and depth-first sampling (DFS). This allows Node2Vec to capture both homophily and structural equivalence, making the learned node embeddings more expressive.

2) *Graph autoencoders*.

- **GAE** (Kipf & Welling, 2016). GAE involves an encoder-decoder architecture, where the encoder is a GCN that transforms node features into latent embeddings by aggregating information from neighboring nodes. The embeddings are then used by the decoder, which typically applies a simple inner product operation to reconstruct the graph structure, such as predicting edges between nodes.
- **VGAE** (Kipf & Welling, 2016). VGAE extends GAE by introducing a probabilistic framework using a variational autoencoder (VAE) setup. It models latent variables with Gaussian distributions, enabling the generation of node embeddings that capture uncertainty. This design improves the model's ability to capture complex structures in graphs, especially in tasks like link prediction.

3) *Graph contrastive methods*.

The mode of GCL has three mainstreams: local-to-local, global-to-global, and global-to-local (Zhu et al., 2021b). A classic example of local-to-local is GRACE (Zhu et al., 2020b), which generates two graph views by augmentations and the same nodes in augmented views are positive while all the other node pairs are negative. Global-to-global mode is often used with multiple graphs in the graph classification task, with GraphCL (You et al., 2020) as an early but influential trial. For the global-to-local perspective, positive pairs are taken as the global representation and nodes of augmented views, and negative pairs are the global representation and nodes of corrupted views. DGI (Velickovic et al., 2019) is a typical example.

- **GRACE** (Zhu et al., 2020b). GRACE generates two graph views by corruption and learns node representations by maximizing the agreement of node representations in these two views. To provide diverse node contexts for the contrastive objective, GRACE proposes a hybrid scheme for generating graph views on both structure and attribute levels.

- **GCA** (Zhu et al., 2021c). GCA proposes adaptive augmentation that incorporates various priors for topological and semantic aspects of the graph. On the topology level, GCA designs augmentation schemes based on node centrality measures, while on the node attribute level, GCA corrupts node features by adding more noise to unimportant node features.

- **DGI** (Velickovic et al., 2019). DGI relies on maximizing mutual information between patch representations and corresponding high-level summaries of graphs—both derived using established graph convolutional network architectures. The learned patch representations summarize subgraphs centered around nodes of interest, and can thus be reused for downstream node-wise learning tasks.

- **MVGRL** (Hassani & Khasahmadi, 2020). MVGRL introduces a self-supervised approach for learning node and graph-level representations by contrasting structural views of graphs. MVGRL shows that contrasting multi-scale encodings does not improve performance, and the best performance is achieved by contrasting encodings from first-order neighbors and graph diffusion.

- **ProGCL** (Xia et al., 2022). ProGCL observes limited benefits when adopting existing hard negative mining techniques of other domains in graph contrastive learning. ProGCL proposes an effective method to estimate the probability of a negative being true and devises two schemes to boost the performance of GCL.

4) *Non-contrastive methods*.

- **CCA-SSG** (Zhang et al., 2021). CCA-SSG optimizes a novel feature-level objective that aligns features across different graph augmentations. It uses decorrelation to prevent degenerate solutions, allowing the model to learn invariant node representations. The model avoids a mutual information estimator or negative samples, which simplifies training and reduces computational complexity.

- **BGRL** (Thakoor et al., 2022). BGRL avoids the use of negative samples by predicting different augmentations of the input graph. BGRL relies on a bootstrapping mechanism, where one branch predicts the output of another branch that is not updated by gradient descent. This method eliminates the complexity of contrastive learning and negative sampling, making it more scalable.

5) *Heterophily baselines*.

- **PolyGCL** (Chen et al., 2024). PolyGCL integrates spectral polynomial filters into graph contrastive learning, enabling it to handle both homophilic and heterophilic graphs. The method generates different spectral views using polynomials and incorporates high-pass information into the contrastive objective.

- **HGRL** (Chen et al., 2022). HGRL introduces self-supervised learning for heterophilic graphs by capturing distant neighbors and preserving original node features. It achieves this through carefully designed pretext tasks optimized via high-order mutual information, avoiding reliance on labels.

- **GraphACL** (Xiao et al., 2024). GraphACL focuses on an asymmetric view of neighboring nodes. The algorithm captures both one-hop local neighborhood information and two-hop monophily similarity, crucial for modeling heterophilic structures.

- **SP-GCL** (Wang et al., 2023). SP-GCL introduces a single-pass graph contrastive learning method without augmentations. It theoretically guarantees performance across both homophilic and heterophilic graphs by studying the concentration property of features obtained through neighborhood propagation.

- **DSSL** (Xiao et al., 2022). DSSL decouples neighborhood semantics in self-supervised learning for node representation. It introduces a latent variable model that decouples node and link generation, making it flexible to different graph structures. The method utilizes variational inference for scalable optimization, improving downstream performance without relying on homophily assumptions.

We categorize the baselines in the **graph classification task** into 1) graph kernel methods including GL (Shervashidze et al., 2009), WL (Shervashidze et al., 2011), and DGK (Yanardag & Vishwanathan, 2015), 2) traditional graph embedding methods including node2vec (Grover & Leskovec, 2016), sub2vec (Adhikari et al., 2018), and graph2vec (Narayanan et al., 2017), 3) contrastive learning methods including InfoGraph (Sun et al., 2020), GraphCL (You et al., 2020), MVGRL (Hassani & Khasahmadi, 2020), JOAOv2 (You et al., 2021), ADGCL (Suresh et al., 2021) as introduced in recent works. The design details are as follows.

1) *Graph kernel methods*.

- **Graphlet Kernel** (GL) (Shervashidze et al., 2009). GL works by counting the number of small subgraphs (known as graphlets) of a fixed size that appear in each graph. The comparison of these counts across graphs allows the kernel to capture the local topological structures of the graphs, making it useful for tasks such as graph classification.

- **Weisfeiler-Lehman Sub-tree Kernel** (WL) (Shervashidze et al., 2011). WL extends the concept of graph kernels by applying the Weisfeiler-Lehman test of isomorphism on graphs. It involves iteratively relabeling the nodes of the graphs based on the labels of their neighbors and then using these relabelings to define a kernel, typically counting matching sub-trees.

- **Deep Graph Kernel** (DGK) (Yanardag & Vishwanathan, 2015). DGK combines deep learning techniques with graph kernels. It first learns a low-dimensional representation of the graphs through unsupervised learning (often using a form of graph embedding or autoencoders), then applies traditional kernel methods to these representations.

2) *Traditional graph embeddings*.

- **Node2Vec** (Grover & Leskovec, 2016). Node2Vec is built on DeepWalk by introducing a flexible biased random walk strategy to explore network neighborhoods. The key innovation is balancing BFS and DFS. This allows Node2Vec to capture both homophily and structural equivalence, making the learned node embeddings more expressive.

- **Sub2Vec** (Adhikari et al., 2018). Inspired by the word2vec model, sub2vec learns vector representations for subgraphs in a graph. It treats each subgraph as a "word" and the entire graph as a "document" to learn embeddings that capture the structural and contextual properties of subgraphs.

- **Graph2Vec** (Narayanan et al., 2017). Similar to sub2vec, graph2vec is designed to learn embeddings for entire graphs. By treating each graph as a "document" and graph substructures as "words," graph2vec employs a document embedding approach to learn a fixed-size vector representation for each graph.

3) *Graph contrastive methods*.

- **GraphCL** (You et al., 2020). GraphCL designs four types of graph augmentations to incorporate various priors and learns graph-level representations by maximizing the global representations of two views for a graph.

- **InfoGraph** (Sun et al., 2020). InfoGraph maximizes the mutual information between the graph-level representation and the representations of substructures of different scales (*e.g.*, nodes, edges, triangles). By doing so, the graph-level representations encode aspects of the data that are shared across different scales of substructures.

- **ADGCL** (Suresh et al., 2021). ADGCL proposes a novel principle, adversarial GCL, which enables GNNs to avoid capturing redundant information during training by optimizing adversarial graph augmentation strategies used in GCL.

- **JOAO** (You et al., 2021). JOAO proposes a unified bi-level optimization framework to automatically, adaptively, and dynamically select data augmentations when performing GraphCL on specific graph data. JOAO is instantiated as min-max optimization.

O.3 SETTINGS

For the node classification task, following Zhu et al. (2020b); Velickovic et al. (2019); Hassani & Khasahmadi (2020), we use linear evaluation protocol, where the model is trained in an unsupervised manner and feeds the learned representation into a linear logistic regression classifier. In the evaluation procedure, we randomly split each dataset with a training ratio of 0.8 and a test ratio of 0.1, and

hyperparameters are fixed the same way for all the experiments. Each experiment is repeated ten times with mean and standard derivation of accuracy score.

For the graph classification task, we use Adam SGD optimizer with the learning rate selected in $\{10^{-3}, 10^{-4}, 10^{-5}\}$ and the number of epochs in $\{20, 100\}$. For PROP, we only search the propagation step $K$ in the range of $[0, 1, 2, 3, 5, 10]$. Following Sun et al. (2020); You et al. (2020), we feed the generated graph embeddings into a linear Support Vector Machine (SVM) classifier, and the parameters of the downstream classifier are independently tuned by cross-validation. The C parameter is tuned in $\{10^{-3}, 10^{-2}, \cdots, 10^2, 10^3\}$. We report the mean 10-fold cross-validation accuracy with standard deviation. All experiments are conducted on a single 24GB NVIDIA GeForce RTX 3090.

### O.4 HYPERPARAMETER

For all methods, we train the linear classifier for 2000 epochs with a learning rate of 0.01 and no weight decay. For hyperparameters of the model architecture and the unsupervised training procedure, we maintain consistency in the hyperparameter search space across methods as much as possible.

Specifically, for GRACE, we search the temperature $\tau$ in [0.1, 0.5, 1.0], the projector hidden dimension in [128, 256, 512], the learning rate in [0.01, 0.001], fix the patience as 50, and all augmentation rates as 0.2. For DGI, we search the learning rate in [0.01, 0.001], the early-stopping patience in [50, 100], and the hidden dimension in [128, 256, 512]. For CCA-SSG, we search the training epochs in [20, 50, 100], $\lambda$ in [1e-3, 5e-4], the hidden dimension in [128, 256, 512], and fix all augmentation ratios as 0.2. For GCA, we search the temperature $\tau$ in [0.1, 0.5, 1.0], the projector hidden dimension in [128, 256, 512], the drop scheme in [pr, degree, evc], and fix the early-stopping patience as 50, the learning rate as 0.01, and all augmentation ratios as 0.2. For BGRL, we search the predictor hidden dimension in [128, 256, 512], the learning rate in [1e-4, 1e-5], the weight decay in [0, 1e-5], fix the learning rate warmup epochs as 1000, the momentum moving as 0.99. For DeepWalk, we search the vector dimension in [128, 256, 512], the context window size in [5, 10], the walk number in [10 20], and the walk length in [40, 80]. For Node2Vec, we search the vector dimension in [128, 256, 512], the walk number in [10 20], the probability $p$ in [0.5, 1.0], $q$ in [0.5, 1.0], and fix the context window size as 10, and the walk length as 80. For MVGRL, we search the learning rate in [0.01, 0.001], the early stopping patience in [50, 100], and the hidden dimension in [128, 256, 512]. For GAE and VGAE, we search the learning rate in [0.01, 0.001], the early stopping patience in [50, 100], and the hidden dimension in [128, 256, 512]. For the heterophily baselines in 6.2, we use the optimal hyperparameter combinations provided in the original papers.

## P  PROOF OF THEOREMS

## Q  PROOF OF THEOREM 4.1

Here we present the proof of Theorem 4.1.

*Proof.* The gradient update of the Dirichlet energy objective (Equation 2) gives the following update rule of node features $\mathbf{H}$,

$$\mathbf{H} - \alpha \frac{\partial \mathcal{L}(\mathbf{H})}{\partial \mathbf{H}} = \mathbf{H} - 2\alpha \hat{\mathbf{L}} \mathbf{H} = ((1 - 2\alpha)\mathbf{I} + 2\alpha \hat{\mathbf{A}})\mathbf{H}, \tag{7}$$

where the $\alpha$ is the step size. When we choose the learning rate $\alpha = 0.5$, we recover the propagation operation in Equation 1, *i.e.*, $\mathbf{H}_{\text{new}} = \hat{\mathbf{A}}\mathbf{H}$.

For convergence analysis, we have

$$\begin{aligned}
\mathcal{L}(\mathbf{H}^{(K)}) &= (\hat{\mathbf{A}}^K \mathbf{H}^{(0)})^\top \hat{\mathbf{L}} (\hat{\mathbf{A}}^K \mathbf{H}^{(0)}) \\
&= \mathbf{H}^{(0)^\top} \hat{\mathbf{A}}^K \hat{\mathbf{L}} \hat{\mathbf{A}}^K \mathbf{H}^{(0)} \\
&= \mathbf{H}^{(0)^\top} (\hat{\mathbf{A}}^{2K} - \hat{\mathbf{A}}^{2K+1}) \mathbf{H}^{(0)}.
\end{aligned} \tag{8}$$

As is known, the range of eigenvalue of $\hat{\mathbf{L}}$ is $[0, 2]$, therefore, the eigenvalues of $\hat{\mathbf{A}}$ belong to $[-1, 1]$. The eigenvalue of $\hat{\mathbf{L}}$ equals 2 if and only if the graph is bipartite. So for non-bipartite graphs, which is often the case for complex graphs in real world, we have the eigenvalues of $\hat{\mathbf{A}}$ belong to $(-1, 1]$. Then when $K$ goes towards infinity, we have $\lim_{K \to +\infty} \mathcal{L}(\mathbf{H}^{(K)}) = 0$, which ends the proof. $\qquad\square$

## R  PROOF OF THEOREM 4.2

Here we present the proof of Theorem 4.2.

*Proof.* A key step is to notice that the alignment objective Equation 3 is closely relevant to the Dirichlet energy when $f(x_i) = \mathbf{H}_i, \forall\, i \in [N]$:

$$\mathcal{L}_{\text{align}}(f) = -\sum_{i,j} \mathbf{A}_{ij}[\mathbf{H}_i^\top \mathbf{H}_j]/(\sum_{i,j} \mathbf{A}_{ij}) = \mathbf{H}^\top \mathbf{A}\mathbf{H}/(\sum_{i,j} \mathbf{A}_{ij}) = \mathbf{H}^\top(\mathbf{I}-\mathbf{L})\mathbf{H}/(\sum_{i,j} \mathbf{A}_{ij}). \quad (9)$$

It is easy to see that graph convolution converges to identical vectors, known as oversmoothing. Therefore, we have $\forall\, i, j, (\mathbf{H}_\infty)_i = (\mathbf{H}_\infty)_j$. Therefore,

$$\lim_{k\to\infty} \mathcal{L}_{\text{align}}(f_k) = \mathbf{H}_\infty^\top \mathbf{A}\mathbf{H}_\infty/(\sum_{i,j} \mathbf{A}_{ij}) = (\sum_{i,j} \mathbf{A}_{ij})/(\sum_{i,j} \mathbf{A}_{ij}) = -1,$$

which concludes the proof. □

