# OpenReview forum: "Propagation Alone is Enough for Graph Contrastive Learning"
_ICLR.cc/2025/Conference — Submitted to ICLR 2025_

### Official Review · Reviewer_xD6G · 2024-10-29

**Soundness:** 3
**Presentation:** 3
**Contribution:** 2
**Rating:** 5
**Confidence:** 4

**Summary:**

This paper makes an interesting observation about graph contrastive learning models. While recent GCL methods have become increasingly complex with sophisticated architectures and designs, the authors demonstrate that simple feature propagation alone can achieve competitive performance. They introduce PROP, a training-free propagation method that simply propagates node features along the graph structure. Despite its simplicity, PROP achieves competitive or even superior performance compared to state-of-the-art GCL methods across various benchmarks, particularly on heterophilic graphs. The authors further deepen their investigation by analyzing existing GCL methods from a decoupling perspective, separating the propagation and transformation components. Their analysis reveals that GCL methods struggle to learn effective transformation weights (performing no better than random), while showing potential in learning propagation coefficients. Based on these insights, they propose PROPGCL, a simplified version that removes transformation layers and only learns propagation coefficients. PROPGCL not only maintains competitive performance but also offers computational advantages.

**Strengths:**

1. The paper effectively shows an interesting fact that simple propagation can work as well as complex GNN encoders in graph contrastive learning.

2. The paper's key strength lies in its systematic experimental design, particularly in Section 5.2. Through well-designed ablation studies comparing fixed propagation coefficients, fixed transformation weights, and an all-one baseline, the authors convincingly show that GCL methods can learn good propagation coefficients but struggle with transformation weights.

3. The experimental evaluation is remarkably comprehensive, comparing against diverse GNN architectures including spectral-based models (ChebNetII), spatial-based methods (GCN, GAT), and polynomial variants (GPRGNN, BernNet). Their simple PROP method consistently matches or outperforms these sophisticated architectures, especially on heterophilic graphs. The authors then show that their PROPGCL variants can further improve upon PROP.

**Weaknesses:**

This paper has a few areas that could be strengthened, primarily in its theoretical analysis.

1. In Section 4.1, where the authors present propagation as a non-parametric learning approach, the theoretical foundation could be more rigorous. The claim that propagation corresponds to gradient descent on Dirichlet energy lacks rigorous mathematical proof, and the cited papers (Zhu et al., 2021a) don't very clearly support this interpretation. The connection between propagation and non-parametric unsupervised learning remains more of an intuitive argument than a mathematically grounded one.

2.  The claim "the propagation operator can also be understood as a special GCL method, where the positive samples are randomly drawn from neighboring nodes" lacks rigorous proof. The authors directly make this connection without formal mathematical justification, and do not explain how propagation's deterministic neighbor aggregation truly corresponds to random sampling of positive pairs in contrastive learning.

3. Theorem 4.1 shows propagation minimizes the alignment loss, this actually corresponds to a mode collapse scenario where all node representations become identical - a situation that contrastive learning typically aims to avoid.

**Questions:**

1. In Section 4.1, the notation \( x^{+} \) appears in the alignment loss equation. Could you clarify what \( x^{+} \) specifically represents in this context? Explicitly defining this and other notations throughout would enhance clarity and help readers follow your ideas more smoothly.

2. Suggestion: The paper currently presumes familiarity with the standard Graph Contrastive Learning (GCL) pipeline. Including a brief overview of the GCL pipeline in the main text would benefit readers who may not have a deep background in GCL.

3. Suggestion: The computational efficiency appears to be a major advantage of PROP/PROPGCL, but this important result is placed in the appendix instead of main context.

---

> ### Author Response · Authors · 2024-11-21
>
> We thank Reviewer xD6G for their kind recognition of our contributions and for raising insightful points, which we address below.
>
> ---
>
> **Q1.**
> In Section 4.1, where the authors present propagation as a non-parametric learning approach, the theoretical foundation could be more rigorous. The claim that propagation corresponds to gradient descent on Dirichlet energy lacks rigorous mathematical proof, and the cited papers (Zhu et al., 2021a) don't very clearly support this interpretation. The connection between propagation and non-parametric unsupervised learning remains more of an intuitive argument than a mathematically grounded one.
>
> **A1.**
> We establish a formal connection between the Dirichlet energy objective and the propagation procedure in **the following theorem**. For reference, we reproduce Equation 1 and Equation 2 from the paper:
> $$\begin{align}
>  & H^{(K+1)} =\hat{A}H^{(K)} \quad (1) \\\\
>  & \mathcal{L}(H)=H^\top \hat{L}H = \sum_{ij}  \hat{A}_{ij}  \Vert H_i- H_j \Vert ^2 \quad (2).
> \end{align}$$
> ***Theorem.*** *For a learning step size of $\alpha=0.5$, the propagation procedure (Equation 1) optimizes the Dirichlet energy objective (Equation 2) and converges to a state where the energy $\mathcal{L}(H^{(K)}) \to 0$ as $K \to +\infty$ for non-bipartite graphs.*
>
> *Proof.* The gradient update of the Dirichlet energy objective (Equation 2) gives the following update rule of node features $H$,
> $$
> H-\alpha\frac{\partial \mathcal{L}(H)}{\partial H} = H-2\alpha\hat{L}H = ((1-2\alpha)I + 2\alpha\hat{A})H,
> $$
> where the $\alpha$ is the step size. By choosing the learning rate $\alpha=0.5$, the update rule simplifies to $H_{new}=\hat{A}H$, which recovers the propagation operation in Equation 1.
> To analyze convergence, we consider the Dirichlet energy objective after $K$ propagation steps:
> $$
> \mathcal{L}(H^{(K)}) = (\hat{A}^KH^{(0)})^\top\hat{L}(\hat{A}^KH^{(0)})
> =H^{{(0)}^\top}\hat{A}^K\hat{L}\hat{A}^KH^{(0)}
> =H^{{(0)}^\top}(\hat{A}^{2K}-\hat{A}^{2K+1})H^{(0)}.
> $$
> The eigenvalues of $\hat{L}$ lie in the range $[0,2]$, which implies the eigenvalues of $\hat{A}$ lie in $[-1,1]$. The eigenvalue of $\hat{L}$  equals 2 if and only if the graph is bipartite. For non-bipartite graphs—common for complex real-world graphs—the eigenvalues of $\hat{A}$  lie in $(-1,1]$. As $K \to +\infty$ , the contribution of higher powers of $\hat{A}$ diminishes, and we have $\lim_{K \to +\infty}\mathcal{L}(H^{(K)})=0$, which ends the proof.
>
> ---
>
> **Q2.**
> The claim "the propagation operator can also be understood as a special GCL method, where the positive samples are randomly drawn from neighboring nodes" lacks rigorous proof. The authors directly make this connection without formal mathematical justification, and do not explain how propagation's deterministic neighbor aggregation truly corresponds to random sampling of positive pairs in contrastive learning.
>
> **A2.**
>
> Using neighboring nodes can be understood as a form of view generation in GCL. Formally, **this involves designing a permutation matrix $P$ that transforms the graph such that $A'=P^\top AP$ and $X'=PX$.** The same row of $X$ (or $A$) and $X'$ (or $A'$) corresponds to neighboring nodes in the original graph. This kind of view generation is also applied in the previous work and shows satisfying experimental performance [1,2].
>
> Consider a simple example of a triangle graph with three nodes $v_1$, $v_2$, and $v_3$, connected as $(v_1,v_2)$, $(v_1,v_3)$ $(v_2,v_3)$. A specific permutation $P=\begin{pmatrix}0 & 1&0\\\\0&0&1\\\\1&0&0\end{pmatrix}$ transforms the original graph's adjacency matrix $A=\begin{pmatrix}0&1&1\\\\1&0&1\\\\1&1&0\end{pmatrix}$, $X=\begin{pmatrix}x_1\\\\ x_2\\\\x_3\end{pmatrix}$ into $A'=P^\top A P=\begin{pmatrix}0&1&1\\\\1&0&1\\\\1&1&0\end{pmatrix}$, $X'=PX=\begin{pmatrix}x_2\\\\ x_3\\\\x_1\end{pmatrix}$. The corresponding nodes in $G=(A,X)$ and $G'=(A', X')$ form positive pairs.
>
> Based on random sampling, other choices of $P$ are possible, such as transforming $X$ to $X'=(x_3, x_1, x_2)^\top$. For node $v_1$, the probabilities of transferring to $v_2$ and $v_3$ are equal.  When the sampling process is repeated sufficiently, the positive samples $(v_1,v_2)$ and $(v_1,v_3)$ are sampled with approximately equal frequency, corresponding to the neighboring set in propagation.
>
> More formally, consider the alignment loss defined in the paper:
>
> $L_{align}(f)=-E_{{x_{i}, x_{j} \sim p(x_i, x_j)}} [f(x_i)^\top f(x_j)].$
> (The expectation $\mathbb{E}$ is written as $E$ for rendering problems of OpenReview.)
>
> Here, the probability distribution $p(x_i, x_j)=A_{ij}/\sum_{i,j}A_{ij}$ is defined as the normalized edge weight between nodes $v_i$ and $v_j$. **When the sampling process is efficient, we can approximate the neighbor sets in propagation as positive pairs.**
>
> [1]. Augmentation-Free Self-Supervised Learning on Graphs. Lee et al., AAAI 2022.
> [2]. Neighbor Contrastive Learning on Learnable Graph Augmentation. Shen et al., AAAI 2023.

---

> ### Author Response · Authors · 2024-11-21
>
> **Q3.**
> Theorem 4.1 shows propagation minimizes the alignment loss, this actually corresponds to a mode collapse scenario where all node representations become identical - a situation that contrastive learning typically aims to avoid.
>
> **A3.**
> When the propagation step $K$ approaches infinity, node representations indeed collapse to identical values, a phenomenon known as over-smoothing. However, **in practice, we limit $K$ to a finite range, typically fewer than five. This ensures that representation collapse does not occur, as supported by our experimental results.**
>
> ---
>
>
> **Q4.**
> In Section 4.1, the notation ( $x^{+}$ ) appears in the alignment loss equation. Could you clarify what ( $x^{+}$ ) specifically represents in this context? Explicitly defining this and other notations throughout would enhance clarity and help readers follow your ideas more smoothly.
>
>  **A4.**
> The notation $(x, x^{+})$ represents positive pairs. To enhance clarity and maintain consistency with the definition of $A_{ij}$,  we have updated the notation in the revised version to $(x_i, x_j)$. The revised alignment loss equation is:
> $$L_{align}(f)=-E_{x_i, x_j \sim p(x_i, x_j)}[f(x_i)^\top f(x_j)],$$
> where $p(x_i, x_j)=A_{ij}/\sum_{i,j}A_{ij}$ denotes the normalized edge weight between nodes $v_i$ and node $v_j$.
>
> ---
>
> **Q5.**
> Suggestion: The paper currently presumes familiarity with the standard Graph Contrastive Learning (GCL) pipeline. Including a brief overview of the GCL pipeline in the main text would benefit readers who may not have a deep background in GCL.
>
> **A5.**
> Thank you for the suggestion.  We have added a brief overview of GCL pipeline in the background section in the revised version.
>
> ---
>
> **Q6.**
> Suggestion: The computational efficiency appears to be a major advantage of PROP/PROPGCL, but this important result is placed in the appendix instead of main context.
>
> **A6.**
> Thank you for this valuable suggestion. In the revised paper, we have moved the efficiency analysis to Section 6.3 of the main body, while the original ablation study on filter choices has been relocated to the appendix.
>
> ---
>
> Thanks for your comments and hope our answers could address your concerns. Please let us know if you have additional questions.

---

> ### Author Response · Authors · 2024-11-25
> **Any further questions?**
>
> Dear Reviewer xD6G,
>
> Thanks for your constructive and valuable comments on our paper. We have tried our best to address your concerns and revised our paper accordingly. Could you please to have a check if there are any unclear points? We are certainly happy to answer your any further questions.

---

> > ### Comment · Reviewer_xD6G · 2024-11-27
> >
> > Thank you for the detailed responses to my concerns. Based on your analysis, the GCL paradigm itself appears less meaningful now - it might be more effective to directly optimize propagation in downstream tasks. While current GCL methods use linear evaluation borrowed from CL, downstream tasks could certainly employ GNNs for evaluation. I maintain my score, and suggest that the paper might be stronger if it focused on arguing how the GCL paradigm fails to improve performance compared to supervised GNNs.

---

> ### Author Response · Authors · 2024-11-28
>
> Thank you for your insightful comments. It is precisely our key contribution to discover (with rigorous evidence) that the *GCL paradigm itself appears less meaningful.* As shown in the paper, **one of our main contributions** is illustrating the inferiority of many existing GCL methods compared to the propagation-only PROP. To the best of our knowledge, this has not been revealed in the literature before. Regarding the evaluation protocol, linear-probing remains the most popular (and dominant) evaluation tool in GCL, and we conduct fair comparisons under this common setting. As you suggest, more alternatives should be adopted in GCL evaluation **based on our work**, and we plan to explore this in future work on advanced evaluation protocols for GCL.
>
> For the suggestion that *focus on arguing how the GCL paradigm fails to improve performance compared to supervised GNNs*. We accordingly revise our paper (**Introduction, Section 5**) by underscoring the findings that GCL learns smooth, nearly Gaussian transformation weights compared to supervised learning, causing a degenerative performance.

---

### Official Review · Reviewer_FGWU · 2024-10-31

**Soundness:** 2
**Presentation:** 2
**Contribution:** 2
**Rating:** 3
**Confidence:** 4

**Summary:**

The paper claims that non-parametric propagation is a special form of GCL and outperforms many existing GCN and GCL methods. The authors observe that GCL can effectively learn optimal propagation parameters, leading them to propose PROPGCL, which focuses solely on learning the propagation coefficients.

**Strengths:**

- Adequate experiments.
- The paper is easy to follow.

**Weaknesses:**

- The authors claim the PROP is a special GCL because it has an alignment operation between the node and its neighbors. However, the GCL is aligned node with itself on another view, this is different. Moreover, the commonly used infoNCE also includes the comparison part of negative samples. Therefore, I can't entirely agree that PROP is a special GCL.

- The PROP without a transformation matrix means that can not transform the node feature from the feature space to the class space. How to make PROP to make the prediction?
- The PROP form has been proposed in many previous works, such as SGC, PMLP[1], etc. What is the difference between PROP and SGC?

- The details of the proposed method should be provided, such as PROP-GRACE.

[1] Graph Neural Networks are Inherently Good Generalizers: Insights by Bridging GNNs and MLPs

**Questions:**

See weaknesses.

**Details Of Ethics Concerns:**

None.

---

> ### Author Response · Authors · 2024-11-21
>
> We thank Reviewer FGWU for the thoughtful feedback and valuable comments on our work.
>
> ---
>
> **Q1.**
> 1). The authors claim the PROP is a special GCL because it has an alignment operation between the node and its neighbors. However, the GCL is aligned node with itself on another view, this is different.
> 2). Moreover, the commonly used infoNCE also includes the comparison part of negative samples. Therefore, I can't entirely agree that PROP is a special GCL.
>
> **A1.**
> View generation is a broad research topic in GCL and is not limited to operations like randomly dropping edges or masking features. **Neighboring nodes can also be considered a type of generated view by applying a specific permutation matrix $P$ to the graph**, *i.e.*, $A'=P^\top AP$, $X'=PX$. This type of neighboring-node view generation has been used in previous works and has demonstrated strong experimental performance [1, 2].
>
> **In contrastive learning, negative samples are not always necessary.** For example, methods such as BGRL [3] and CCA-SSG [4] exclude negative samples for efficiency. In InfoNCE, negative samples are typically introduced to prevent representation collapse when aligning positive pairs. However, **in PROP, the alignment occurs over a finite number of propagation steps (usually fewer than 5 in practice), which inherently avoids collapse.** Therefore, there is no need to include negative samples in this case.
>
> [1]. Augmentation-Free Self-Supervised Learning on Graphs. Lee et al., AAAI 2022.
> [2]. Neighbor Contrastive Learning on Learnable Graph Augmentation. Shen et al., AAAI 2023.
> [3]. Large-scale Representation Learning on Graphs via Bootstrapping. Thakoor et al., ICLR 2022.
> [4]. From Canonical Correlation Analysis to Self-Supervised Graph Neural Networks. Zhang et al., NeurIPS 2021.
>
> ---
>
> **Q2.**
> The PROP without a transformation matrix means that can not transform the node feature from the feature space to the class space. How to make PROP to make the prediction?
>
> **A2.**
> GCL methods typically involve two stages: pretraining and evaluation. During the pretraining stage, node representations are learned using self-supervised signals without access to downstream labels. In the evaluation stage, a linear classifier is trained in a supervised manner to make predictions for downstream tasks. **The projection from the representation space to the class space occurs during the evaluation stage** and follows the same procedure across different GCL methods to ensure fair comparison.
>
> For PROP, node features are propagated during pretraining to generate node representations. **In the evaluation stage, these representations are then input into a linear classifier, which is trained to make predictions.**
>
> In the revised version of the paper, we have added a detailed description of the GCL pipeline to the background section for greater clarity.
>
> ---

---

> > ### Comment · Reviewer_FGWU · 2024-11-22
> >
> > I don’t quite understand. According to the author, can all low-pass filtering GCNs be called GCL? because they can align the representations of nodes and neighbors?

---

> > > ### Author Response · Authors · 2024-11-22
> > >
> > > Thank you for your insightful comments. We want to further clarify that we were not trying to regard all (low-pass) GCNs as GCL, but to point out that there is a fundamental connection between GCN propagation and GCL (Section 4.1) ---  such that **propagation** could attain similar performance to GCL even without training. Indeed, GCN's **transformation** stage cannot be equated with some form of contrastive learning, evidenced by the poor expressive power of untrained GCNs in unsupervised settings [1].
> > >
> > > We have further made this clearer in the revision (**Section 4.1**) through an elaboration. Thank you for bringing it up to us, and please let us know if there is more to clarify.
> > >
> > > ---
> > >
> > > [1]. Susheel, et al. Adversarial graph augmentation to improve graph contrastive learning. In NeurIPS, 2021.

---

> ### Author Response · Authors · 2024-11-21
>
> **Q3.**
> The PROP form has been proposed in many previous works, such as SGC, PMLP[1], etc. What is the difference between PROP and SGC?
>
> [1] Graph Neural Networks are Inherently Good Generalizers: Insights by Bridging GNNs and MLPs
>
> **A3.**
> SGC was primarily introduced and discussed in the context of supervised learning. In the GCL framework, **SGC is often used as an encoder with the formulation** $H_{\rm SGC}=\hat{A}^KXW$, where $W$ represents trainable weights. In contrast, **PROP removes the trainable weights entirely, resulting in the formulation** $H_{\rm PROP}=\hat{A}^KX$.
>
> **This distinction is crucial** because our findings (Section 5.1) reveal that the weights $W$ learned during GCL pretraining tend to be overly smoothing and less informative, which can harm downstream performance (Section 4.2).
>
> We emphasize that **our contribution is not focused on introducing PROP as a new formulation**, but rather on investigating the role of different components in existing GCL methods. Through comprehensive experiments, we demonstrate the strong competitiveness of the propagation-only baseline for node classification tasks. Despite its simplicity, the baseline has been largely overlooked in the GCL literature, and our work aims to shed light on its effectiveness.
>
> ---
>
> **Q4.**
> The details of the proposed method should be provided, such as PROP-GRACE.
>
>
> **A4.**
> We have revised the overview of PROPGCL in Section 6.1 and provided detailed explanations of PROP-GRACE in the **baselines** section of Section 6.2.
>
> ----
> We hope our responses address your concerns, and we are happy to provide further clarification if needed.

---

> > ### Comment · Reviewer_FGWU · 2024-11-22
> >
> > For A3, the author's reply does not convince me. SGC can be regarded as first aggregating neighbors to get representation (i.e., an alignment operation between the node and its neighbors) and then using MLP as a classifier for downstream tasks. It also has no parameters when getting the representation. There are many GNNs like this.

---

> > > ### Author Response · Authors · 2024-11-22
> > >
> > > Indeed, as you pointed out, SGC's propagation can be understood as a "representation learning" phase. And this is **exactly the point of our work** -- we want to argue that the significance of this naive baseline is overlooked in existing graph SSL literature, and we justify it by showing that when properly configured, this propagation alone method can perform on par with complex learning-based methods. In other words, **the main focus of this work is not to propose a new alternative method as "PROP", just to have a critical understanding of existing GCL methods through an ablated baseline "PROP"**. The naming as "PROP" is to make this point clear instead of tying the propagation to a special method (as SGC), which **may lead to some confusion since SGC used in GCL literature commonly contains the weight $W$ in practice** [1,2,3].
> > >
> > > We have better clarified our purpose and the connection to SGC in **Method part of Section 4.2** to avoid potential confusion. Thank you for this insightful comments and we hope our revision could address your concern.
> > >
> > > ---
> > >
> > > [1]. Li, et al. Graph Contrastive Representation Learning with Input-Aware and Cluster-Aware Regularization. In Joint European Conference on Machine Learning and Knowledge Discovery in Databases, 2023.
> > >
> > > [2]. Gao, et al. Rethinking Graph Contrastive Learning: An Efficient Single-View Approach via Instance Discrimination. In IEEE Transactions on Multimedia, 2023.
> > >
> > > [3]. Chen, et al. Attribute and structure preserving graph contrastive learning. In AAAI, 2023.

---

> ### Author Response · Authors · 2024-11-25
> **Any further questions?**
>
> Dear Reviewer FGWU,
>
> Thanks for your constructive and detailed comments on our paper. We have tried our best to address your concerns and revised our paper accordingly. Could you please have a check if there are any unclear points? We are certainly happy to answer your any further questions.

---

### Official Review · Reviewer_H333 · 2024-11-02

**Soundness:** 2
**Presentation:** 3
**Contribution:** 2
**Rating:** 6
**Confidence:** 4

**Summary:**

This paper focuses on graph contrastive learning (GCL). The authors reveal that a simple training-free propagation method PROP can achieve competitive results over dedicatedly designed GCL methods.
To further study the underlying reasons, they decouple the propagation and transformation phases of graph neural networks in GCL and find that GCL inadequately learns effective transformation weights while exhibiting potential for solid propagation learning. Motivated by this, the authors proposed a novel GCL method termed PROPGCL which demonstrates its effectiveness on various datasets.

**Strengths:**

1. This paper is well motivated by detailed observation and conducts a thorough analysis.
2. The proposed method is well supported by theoretical findings and previous works. The findings refresh the customary viewpoints on designing GCL methods.
3. The proposed method is simple but shows outstanding performance compared to previous GCL and GSSL methods.

**Weaknesses:**

1. The proposed method is only verified on node-level tasks with strong feature-connection relationships (i.e., strong homophily and heterophily). No study on the more complicated scenario, for example:
   - complex relationship between node features and edges, e.g., partial multidimensional homophily (Block & Grund, 2014)
   - graph structures introduce extra information instead of hinting homophily/heterophily, e.g., superpixel datasets (CIFAR10, MNIST) in Dwivedi et al. (2022).

Without this, the work can only show that previous GCL methods cannot adequately learn feature transformation. However, it cannot demonstrate that learning feature transformation is unnecessary or harmful if we can manage to learn adequately. A further study on it might lead to a stronger statement on the proposed method.

2. There is no evaluation of the inductive setting, including the single-graph datasets (e.g., datasets in Hamilton et al. (2017))
  and multiple-graph datasets (i.e., a dataset containing multiple graphs, such as those in Dwivedi et al. (2022)).













----

- Block, P., & Grund, T. (2014). Multidimensional Homophily in Friendship Networks*. _Network Science_, _2_(2), 189–212.
- Dwivedi, V. P., Joshi, C. K., Laurent, T., Bengio, Y., & Bresson, X. (2022). Benchmarking Graph Neural Networks. _Journal of Machine Learning Research_.
- Hamilton, W., Ying, Z., & Leskovec, J. (2017). Inductive Representation Learning on Large Graphs. _Adv. Neural Inf. Process. Syst._, _30_.

**Questions:**

1. Is $\mathbf{A}$ including self-loops? If not, $\mathbf{A}^K$ or $\hat{\mathbf{A}}$ might not include all $K$-hops neighboring nodes as described in the text.

2. Is it possible to extend the PROP-DGI to multiple graph settings, such as the datasets in Dwivedi et al. (2022)? Any specific reasons that PROP-DGI is only evaluated in transductive setting?



- Dwivedi, V. P., Joshi, C. K., Laurent, T., Bengio, Y., & Bresson, X. (2022). Benchmarking Graph Neural Networks. _Journal of Machine Learning Research_.

---

> ### Author Response · Authors · 2024-11-21
>
> We thank Reviewer H333 for their careful reading and for appreciating the thorough analysis and impactful findings of our work. Below, we summarize and address your main concerns.
>
> ----
>
> **Q1.**
> The proposed method is only verified on node-level tasks with strong feature-connection relationships (i.e., strong homophily and heterophily). No study on the more complicated scenario, for example:
> -  complex relationship between node features and edges, e.g., partial multidimensional homophily (Block & Grund, 2014)
> - graph structures introduce extra information instead of hinting homophily/heterophily, e.g., superpixel datasets (CIFAR10, MNIST) in Dwivedi et al. (2022).
>
> Without this, the work can only show that previous GCL methods cannot adequately learn feature transformation. However, it cannot demonstrate that learning feature transformation is unnecessary or harmful if we can manage to learn adequately. A further study on it might lead to a stronger statement on the proposed method.
>
> **A1.**
> For datasets with multidimensional homophily, we could not locate the dataset mentioned in Block & Grund (2014). If you know where to access this dataset, we would be happy to conduct additional experiments and include the results in our study.
>
> For superpixel datasets (CIFAR10, MNIST) from Dwivedi et al. (2022), we note that these are **graph classification benchmarks** with **less informative node features** (e.g., CIFAR10 has 3-dimensional node features, and MNIST has 5-dimensional node features). We evaluate PROP on similar graph classification tasks (**Appendix A**), where graph structures introduce extra information instead of hinting homophily/heterophily.  Across datasets, the mean performance gap between PROP and the best-performing methods is 2.82%. **Considering PROP’s training-free advantage, we believe these results are still noteworthy.** We hypothesize that the less competitive performance is due to the limited or even null node features typically found in common graph classification benchmarks, which hinder PROP's effectiveness. **In the revised paper, we have narrowed the scope of our study to GCL for node classification tasks and will explore graph classification scenarios in future work**.
>
> Our work highlights a critical issue in **existing** GCL methods: the inadequate learning of transformation weights, which has been largely overlooked in the literature. We address this problem with a simple yet effective solution—removing the transformation stage entirely—demonstrating surprising effectiveness. **While we acknowledge that carefully designed methods or advanced contrastive principles may eventually overcome these limitations, our study makes a meaningful contribution by exposing this overlooked problem and opening new avenues for developing more effective GCL methods.** To ensure clarity and avoid overstatement, we have carefully revised the paper to refine our claims and contributions.
>
> ---
>
> **Q2.**
>  There is no evaluation of the inductive setting, including the single-graph datasets (e.g., datasets in Hamilton et al. (2017)) and multiple-graph datasets (i.e., a dataset containing multiple graphs, such as those in Dwivedi et al. (2022)).
>
> **A2.**
> We conducted experiments in the inductive setting on the single-graph dataset **Reddit** and the multiple-graph dataset **PPI**. The experimental settings, including data splitting and training hyperparameters, follow those in Hamilton et al. (2017). The results are summarized in the table below:
>
> -   For PPI (a multi-graph benchmark with 50-dimensional node features), PROP (K=2) achieves an F1 score of **0.7527**, which is comparable to GRACE's score of **0.7548**.
> -   For Reddit, PROP (K=2) achieves an F1 score of **0.8452**, outperforming GRACE which achieves **0.8185**.
>
> These results validate the effectiveness of PROP in node classification tasks under the inductive setting.
>
> **F1 score comparison on PPI**
> |  Method      |F1 score |
> |--------------|---------|
> | GRACE        | 0.7548  | # $\pm$ 0.0822
> | PROP ($K=0$) | 0.7076  | # $\pm$ 0.0943
> | PROP ($K=1$) | 0.7493  | # $\pm$ 0.0876
> | PROP ($K=2$) | 0.7527  | # $\pm$ 0.0833
>
> **F1 score comparison on Reddit**
> |  Method      |F1 score|
> |--------------|--------|
> | GRACE        | 0.8185 |
> | PROP ($K=0$) | 0.5852 |
> | PROP ($K=1$) | 0.8457 |
> | PROP ($K=2$) | 0.8452 |
>
>
>
> ---

---

> ### Author Response · Authors · 2024-11-21
>
> **Q3.**
> Is  A  including self-loops? If not,  $A^K$ or  $\hat{A}$  might not include all  K-hops neighboring nodes as described in the text.
>
> **A3.**
> Yes, $A$ includes self-loops in practice. Specifically, we use the normalization approach from GCN, where $A'=A+I$, $\hat{A}=D'^{-\frac{1}{2}}A'D'^{-\frac{1}{2}}$. For further details, please refer to the implementation in `prop.py`. We have revised the paper to provide a more precise description.
>
> ---
>
> **Q4.**
> Is it possible to extend the PROP-DGI to multiple graph settings, such as the datasets in [1]? Any specific reasons that PROP-DGI is only evaluated in transductive setting?
>
> [1] Benchmarking Graph Neural Networks. Dwivedi et. al. 2022
>
> **A4.**
> PROP-DGI builds on PROP by introducing learnable propagation through spectral filters. However, **the spectral characteristics of graphs can vary significantly, making it challenging for a filter learned on one graph to generalize effectively to another.** This limitation also explains why spectral GNNs like BernNet, GPRGNN, and ChebNetII are primarily applied to single-graph tasks. We made preliminary attempts to apply PROP-DGI in inductive settings with multiple-graph datasets, but the results were not satisfactory. **We have discussed this limitation in the revised paper’s Limitation Section, acknowledging the need for further exploration to adapt PROP-DGI to such settings.**
>
> ---
>
> Hope our answers could address your concerns. Please let us know if you have additional questions.

---

> > ### Comment · Reviewer_H333 · 2024-11-24
> >
> > Thank you very much for the rebuttal.
> >
> > I believe the rebuttal has addressed most of my concerns.
> > I will raise the score to 6 accordingly.

---

### Official Review · Reviewer_3pXN · 2024-11-03

**Soundness:** 2
**Presentation:** 4
**Contribution:** 2
**Rating:** 5
**Confidence:** 5

**Summary:**

The paper investigates a training free, non-parametric method called PROP for graph contrastive learning. "PROP" is a method which aggregates node features within K-hop neighbors, without any trainable weights. After such a feature propagation phase, the aggregated node features are fed to a downstream classifier for evaluation on node classification / graph classification tasks. The paper claims that for unsupervised learning, only performing "PROP" is enough and is in fact comparable in performance to other learnable methods from literature that utilize sophisticated contrastive objectives. Given the strong performance of "PROP" and claim that propagation of features is sufficient for GCL, the authors further propose "PROPGCL" paradigm, where encoders of traditional GCL methods are replaced with trainable propagation only polynomial graph neural networks. The authors then experimentally show how "PROPGCL" provides performance gains compared to standard encoder choices in said traditional GCL methods.

**Strengths:**

1) Viewing polynomial GNNs (propagation) as performing graph contrastive learning sheds new light into our understanding of contrastive learning methods for graphs.

2) Interesting insight via experiments that show feature transformations are not necessary in certain conditions for effective graph contrastive learning.

3) The paper provides a comprehensive set of experiments, with detailed analysis.

**Weaknesses:**

1) While the experiments are well done, the main claim is not fully supported / backed up. For e.g., in the graph level tasks (Appendix A), the proposed paradigm of only "PROP" is inferior to other GCL focused methods from literature and performance is not comparable. So why is it effective only for node level tasks?

2) The heterophilic graph datasets chosen in the paper are known to be problematic, choosing recent heterophilic datasets (for e.g.., from https://arxiv.org/abs/2302.11640) will provide a more robust argument. IMO the chosen datasets for node level tasks are quite small and limited to meaningfully test GCL in a realistic setting. Investigating the proposed paradigm of "PROP" and "PROPGCL" on large scale graphs from OGB (Open graph benchmark) will provide a more convincing argument.

3)  The novelty of the proposed paradigm of "PROP" is quite limited since it is well known to the community that feature aggregation is a powerful inductive bias that GNNs possess. In fact, many GCL methods commonly employ randomly initialized and untrained GNNs as a baseline [c.f. ADGCL (https://arxiv.org/pdf/2106.05819) See for e.g. RU-GIN.]. Moreover, on node level tasks in heterophilic datasets, it is well known that such feature aggregation property of GNNs is very important.

4) While the authors acknowledge that in cases where there are low quality node features (sparse / noisy / absent), propagation alone can be insufficient. This IMO is one of the biggest drawbacks of the proposed approach. Previous methods aim to deal with such cases.

5) There is very little discussions on the results in conclusion section to string together different experiments into a cohesive argument. IMO the readers will appreciate a concise explanation rather than digging through the appendix.

**Questions:**

1) While the authors cite the paper "A critical look at the evaluation of GNNs under heterophily: Are we really making progress?" Platonov e.t. al, ICLR 2023 (https://arxiv.org/abs/2302.11640) why are the newer datasets not chosen for heterophilic datasets ?

2) Why is PROPGCL not evaluated for graph level tasks.?

3) How is PROP setup for graph level tasks?. Are resulting node feature after feature aggregation pooled into a single feature for downstream classification?

-------
I raise my score from 3-> 5, given that the authors have answered some of my questions and concerns.

---

> ### Author Response · Authors · 2024-11-21
>
> We thank Reviewer 3pXN for careful reading and constructive comments. We summarize and address your main concerns in the following.
>
> ---
>
> **Q1.**
> While the experiments are well done, the main claim is not fully supported / backed up. For e.g., in the graph level tasks (Appendix A), the proposed paradigm of only "PROP" is inferior to other GCL focused methods from literature and performance is not comparable. So why is it effective only for node level tasks?
>
> **A1.**
> As shown in Appendix Table 9, for graph classification tasks, the mean performance gap between PROP and the best methods across datasets is 2.82% (as comparison, with the best gap among methods being 1.52%). Considering PROP’s training-free advantage, we believe these results are still noteworthy. However, PROP does not surpass classical graph SSL baselines in graph classification tasks. We hypothesize that this limitation arises because common graph classification benchmarks often have less informative or even null node features, which likely hinder PROP's effectiveness.
>
> Our theoretical analysis of PROP’s effectiveness is primarily based on the connections between nodes within a single graph (Section 4.1), which aligns closely with node classification tasks. **In the revised version, we have explicitly narrowed the scope of our study to GCL for node classification tasks and clarified that further research will explore the application of PROP to graph classification tasks.**
>
> ---
>
> **Q2.**
>  The heterophilic graph datasets chosen in the paper are known to be problematic, choosing recent heterophilic datasets (for e.g., from  [1] will provide a more robust argument. IMO the chosen datasets for node level tasks are quite small and limited to meaningfully test GCL in a realistic setting.
>
> Investigating the proposed paradigm of "PROP" and "PROPGCL" on large scale graphs from OGB (Open graph benchmark) will provide a more convincing argument.
>
> [1]. A critical look at the evaluation of GNNs under heterophily: Are we really making progress. Platonov et,al. 2023.
>
> **A2.**
> We acknowledge the reviewer's concern and appreciate the suggestion. We have already shown results of PROP and PROPGCL on the benchmark proposed by [1] in **Appendix Table 19**.  As is observed, PROP-DGI achieves the best results in 2 out of 5 benchmarks and attains an average performance of 70.22\%, second only to PolyGCL’s 71.68\%. Notably, PolyGCL is specifically optimized for heterophily graphs, whereas PROP-DGI builds on a simpler DGI framework.
>
> Currently the datasets in the main paper are still the widely used with recent citations. That's why we report them in the main paper and put others in appendix for comprehensive study. **We understand that locating these results in the appendix may be less direct for readers. In the revised version, we moved these benchmark results to the main body for clarity and emphasis.**
>
> Additionally, we have evaluated PROPGCL on the **ogbn-arxiv** dataset, with results presented in **Table 4**. To strengthen our analysis, we further conduct experiments on the **ogbn-products** dataset, which includes 2,449,029 nodes and 61,859,140 edges. The results are as follows:
>
> |  Method  |    Val Acc   |   Test Acc   |
> |----------| ------------ | -------------|
> |    GGD   | 90.59 $\pm$ 0.06 | 75.49 $\pm$ 0.19 |
> | PROP-GGD | 87.88 $\pm$ 0.15 | 73.57 $\pm$ 0.15 |
>
> Due to time constraints, we did not perform hyperparameter tuning for PROP-GGD. Despite this, PROP-GGD achieves competitive performance, with a test accuracy of 73.57% compared to the baseline GGD's 75.49%. Moreover, by removing the transformation stage, PROP-GGD provides significant advantages in computational efficiency and memory consumption. We will refine these experiments further and present the complete results and analysis in the revised version of our paper.
>
> ---

---

> > ### Author Response · Authors · 2024-11-21
> >
> > **Q3.**
> > The novelty of the proposed paradigm of "PROP" is quite limited since it is well known to the community that feature aggregation is a powerful inductive bias that GNNs possess. In fact, many GCL methods commonly employ randomly initialized and untrained GNNs as a baseline [c.f. ADGCL ([https://arxiv.org/pdf/2106.05819](https://arxiv.org/pdf/2106.05819)) See for e.g. RU-GIN.]. Moreover, on node level tasks in heterophilic datasets, it is well known that such feature aggregation property of GNNs is very important.
> >
> > **A3.**
> > We acknowledge the reviewer's observation and clarify that our contribution is **not centered on the novelty of the PROP formulation itself**. PROP can be approximately viewed as a degenerated version of the linear graph convolution layer ($X' = AXW$), where the weights $W$ are removed. **The key difference between PROP and randomly initialized GNNs** (e.g., RU-GIN) is that PROP has no trainable weights, not even randomized ones. In essence, PROP operates purely as a propagation mechanism.
> >
> > Through comprehensive experiments, we demonstrate the strong competitiveness of PROP in node classification tasks. **As far as we know, this highly simple yet competitive "propagation-only" baseline has been largely overlooked in the GCL literature.** For comparison, randomly initialized GNNs (such as RU-GIN) are also free of training but typically show significantly inferior performance, as evidenced by Table 1 of ADGCL.
> >
> > More importantly, the results from PROP raise critical questions about the effectiveness of learned transformation weights in the GCL pre-training stage. As discussed in Section 5.1, we find that the transformation weights learned by some GCL methods often present smooth characteristics and carry limited informativeness. **These observations suggest the need for a deeper rethinking of GCL design principles, particularly regarding the role of learned weights in GCL**.
> >
> > **In conclusion**, while feature aggregation as an inductive bias is well known, our work highlights the overlooked strength of a propagation-only mechanism like PROP in the GCL paradigm . PROP reveals the limitations of transformation weights learned during GCL, offering a valuable perspective for the community to revisit core assumptions in GCL design.
> >
> > ---
> >
> > **Q4.**
> > While the authors acknowledge that in cases where there are low quality node features (sparse / noisy / absent), propagation alone can be insufficient. This IMO is one of the biggest drawbacks of the proposed approach. Previous methods aim to deal with such cases.
> >
> > **A4.**
> > We acknowledge this limitation, which is discussed in the Limitation Section, **particularly in the context of graph classification tasks**. However, **on diverse node classification benchmarks commonly used in mainstream GCL literature, PROPGCL consistently delivers highly competitive performance**. These results validate the effectiveness of our approach in scenarios where node features are informative, as is typical in many benchmark datasets.
> >
> > ----
> >
> > **Q5.**
> > There is very little discussions on the results in conclusion section to string together different experiments into a cohesive argument. IMO the readers will appreciate a concise explanation rather than digging through the appendix.
> >
> >
> > **A5.**
> > We appreciate this valuable feedback. In the revised version of our paper, **we have expanded the conclusion section** to provide a clearer discussion of the results, synthesizing findings from various experiments into a cohesive narrative.
> >
> > ---
> >
> >
> > **Q6.**
> > While the authors cite the paper "A critical look at the evaluation of GNNs under heterophily: Are we really making progress?" Platonov e.t. al, ICLR 2023 ([https://arxiv.org/abs/2302.11640](https://arxiv.org/abs/2302.11640)) why are the newer datasets not chosen for heterophilic datasets ?
> >
> > **A6.**
> > We have evaluated our methods on the benchmark proposed in the cited paper, and the results are presented in **Appendix Table 19**. However, we acknowledge that these results may not be immediately accessible to readers. In the revised version, **we moved the benchmark results to the main body of the paper for ease of reference**.
> >
> > ---

---

> ### Author Response · Authors · 2024-11-21
>
> **Q7.**
>  Why is PROPGCL not evaluated for graph level tasks.?
>
> **A7.**
> PROPGCL enhances PROP by replacing uniform propagation with learnable propagation through spectral filters. However, **the spectral characteristics of graphs can vary significantly, meaning that a filter learned on one graph may not generalize well to another**. This limitation is also why spectral GNNs, such as BernNet, GPRGNN, and ChebNetII, are predominantly applied to node-level tasks.
>
> To align with these characteristics and to ensure the applicability of our method, **we revised the paper to focus on node classification tasks, where PROPGCL demonstrates strong performance**.
>
> ----
>
> **Q8.**
> How is PROP setup for graph level tasks?. Are resulting node feature after feature aggregation pooled into a single feature for downstream classification?
>
> **A8.**
>
> Yes, we use mean pooling of node features to obtain the global graph representation. Specifically, this is implemented as:
>
> `x = scatter(x, data.batch, dim=0, reduce="mean")`
>
> For further details, please refer to the `prop.py` implementation.
>
> In the revised paper, we have added a detailed explanation of the PROP setup for graph-level tasks to ensure clarity.
>
> ---
> Thanks for your comments and hope our answers could address your concerns. Please let us know if you have additional questions.

---

> ### Author Response · Authors · 2024-11-25
> **Any further questions?**
>
> Dear Reviewer 3pXN,
>
> Thanks for your constructive and valuable comments on our paper. We have tried our best to address your concerns and revised our paper accordingly. Could you please to have a check if there are any unclear points? We are certainly happy to answer your any further questions.

---

> > ### Comment · Reviewer_3pXN · 2024-11-29
> > **Reply to authors**
> >
> > I thank the authors for their comments and incorporating many of the suggestions. Aligning the paper on node classification brings more clarity, given the sheer number of experiments. However, it also hurts the applicability and extension of the insights to the graph classification tasks. Moreover, I am still not convinced fully how the proposed propagation only paradigm can be useful in situations especially when we have low-quality node features (sparse / noisy / absent) influence node level tasks more so than graph level tasks.
> >
> > Since many of my questions were answered and given the paper has increased readability, I raise my score accordingly.

---

### Official Review · Reviewer_qb5w · 2024-11-04

**Soundness:** 3
**Presentation:** 3
**Contribution:** 3
**Rating:** 6
**Confidence:** 4

**Summary:**

In this paper, the authors investigate how the propagation and transformation components affect the Graph Contrastive Learning (GCL) methods. They first illustrate that simple uniform propagation can achieve competitive performance compared with GCL baselines through empirical experiments. Then, they further conduct experiments on how the propagation and transformation layers contribute to the GCL process. The results suggest that the GCL process could not help learn very meaningful transformation parameters but could be helpful in learning propagation hyper-parameters. Based on these observations, they further proposed a GCL framework named PROGCL, which adopts different training-free PROP as backbones. Specifically, they replace the encoder in GCL methods with propagation-only polynomial GNNs. The effectiveness of these methods are verified by empirical experiments.

**Strengths:**

1. Originality: The paper majorly investigates how the components in GCL methods work. The findings in this paper are quite novel.

2. Quality: The paper seems to be quite sound. Empirical experiments support the conclusions of this paper well.

3. Clarity: In general, the paper is well written. All the important points and observations are clearly presented. The paper is also well-organized such that different investigations are quite logically presented.

4. Significance: The outcomes from this paper could be significant as they could help provide some guidance on designing effective and efficient GCL methods.

**Weaknesses:**

1. The title and narrative in this paper have been talking about GCL in general. However, this paper is more about GCL for node classification tasks only. Will these conclusions in the paper hold true for other tasks, especially graph classification tasks? It would be better if the authors could conduct further investigations. If not, it might be better if the authors could narrow their scope to GCL for node classification tasks.

2. It would be more convincing if the authors could conduct experiments on larger datasets in addition to the adopted ogbn-arxiv dataset.

**Questions:**

See the questions in the Weakness part.

---

> ### Author Response · Authors · 2024-11-21
>
> We thank Reviewer qb5w for carefully reading and appreciating the novelty and significance of our work on the designs in GCL. Below, we summarize and address your main concerns.
>
> ----
>
> **Q1.**
> The title and narrative in this paper have been talking about GCL in general. However, this paper is more about GCL for node classification tasks only. Will these conclusions in the paper hold true for other tasks, especially graph classification tasks? It would be better if the authors could conduct further investigations. If not, it might be better if the authors could narrow their scope to GCL for node classification tasks.
>
> **A1.**
> We present evaluation results on graph classification tasks in Appendix Table 9. Across datasets, the mean performance gap between PROP and the best-performing methods is 2.82% (as comparison, the best gap among methods being 1.52%). **Considering PROP’s training-free advantage, we believe these results are still noteworthy.** PROP's performance on graph classification tasks is relatively less competitive compared to its strong results on node classification tasks. We hypothesize that this is due to the limited or even null node features typically found in common graph classification benchmarks, which hinder PROP's effectiveness.
>
> The methods and theories proposed in our paper primarily focus on connections between nodes within a single graph, which aligns with node classification tasks. While PROP demonstrates some promise in graph classification, its potential in this area warrants further investigation. **Follow your suggestion, we have narrowed the scope of our study to GCL for node classification tasks and included a clear statement that graph classification will be explored in future work.**
>
> ---
>
> **Q2.**
> It would be more convincing if the authors could conduct experiments on larger datasets in addition to the adopted ogbn-arxiv dataset.
>
> **A2.**
> We conducted additional experiments on the **ogbn-products** dataset, which consists of 2,449,029 nodes and 61,859,140 edges. The results are summarized below:
> |Method| Val Acc |Test Acc|
> |---| ------- | -----------------|
> | GGD | 90.59 $\pm$ 0.06 | 75.49 $\pm$ 0.19 |
> | PROPGGD | 87.88 $\pm$ 0.15| 73.57 $\pm$ 0.15 |
>
> Due to time constraints, we did not perform hyperparameter tuning for PROP-GGD. Despite this, it achieves a satisfying performance with a test accuracy of 73.57%, compared to the baseline GGD's 75.49%. Additionally, by eliminating the transformation stage, PROP-GGD offers significant advantages in terms of computational efficiency and memory consumption. We will further refine this experiment and include detailed results and analysis in the revised version of our paper.
>
> ----
> Thanks for your comments and hope our answers could address your concerns. Please let us know if there is more to clarify. We are happy to address them during the discussion stage.

---

> ### Author Response · Authors · 2024-11-25
> **Any further questions?**
>
> Dear Reviewer qb5w,
>
> Thanks for your constructive and encouraging comments on our paper. We have tried our best to address your concerns and revised our paper accordingly. Could you please have a check if there are any unclear points? We are certainly happy to answer any further questions.

---

> > ### Comment · Reviewer_qb5w · 2024-11-25
> > **Official Response from Reviewer qb5w**
> >
> > Thank you for your efforts in answering the questions. The responses and the updates in the manuscript addressed our concerns. We would like to keep our positive score.

---

### Author Response · Authors · 2024-11-21
**Summary of Paper Updates**

We thank all reviewers for their constructive comments. We have updated the paper accordingly with the following major changes:

-  **Scope Refinement.** Narrow the study's focus to **GCL for node classification tasks**, explicitly stating that future work will explore graph classification scenarios.
- **New Experiments on Larger Datasets.** Add results for the **ogbn-products** dataset (Table 4), demonstrating the scalability and computational efficiency of PROPGCL on large graphs.
- **Enhanced Theoretical Foundation.** Include a formal theorem (Theorem 4.1) connecting the propagation procedure to the Dirichlet energy objective.
-  **Improved Accessibility of Results.** 1). Move the efficiency analysis to Section 6.3 to highlight PROP/PROPGCL's computational advantages. 2). Relocated the results for newer heterophilic benchmarks from the appendix to the main body (Table 7) for better accessibility.
- **Others.**  1). Add a detailed description of the GCL pipeline to the background section. 2). Provide detailed explanations of PROP-GRACE in Section 6.2. 3).  Revise the conclusion section to provide a cohesive discussion of results.

---

### Meta-Review · Area_Chair_VWXT · 2024-12-19

**Metareview:**

This work provides an analysis of Graph Contrastive Learning (GCL) methods, revealing that simple training-free propagation can achieve competitive results compared to state-of-the-art GCL approaches. The authors' investigation into the decoupled propagation and transformation phases of GCL yields possible insights. Building on their analysis, they propose PROPGCL, a GCL framework that replaces traditional encoders with trainable propagation-only polynomial graph neural networks. The empirical results demonstrate the effectiveness and computational advantages of PROPGCL across small datasets in node classification tasks.

The reviewers find the observations interesting and the proposal of PROPGCL well-motivated. In response to a request for clarification on the connection between propagation and non-parametric unsupervised learning, the authors provide a theorem in the rebuttal that establishes a formal link between the Dirichlet energy objective and the propagation procedure.

However, some concerns and questions remain. Several reviewers express skepticism about the sufficiency of propagation (PROP) for unsupervised learning, citing inferior performance on graph-level tasks and limitations in handling low-quality node features. The authors acknowledge these limitations, which suggests that further research is needed to fully explore the potential of PROPGCL. Additionally, there are questions regarding the inductiveness of the method.

Some reviewers also raise a point about whether non-parametric propagation (PROP) can be considered a special form of Graph Contrastive Learning (GCL), as it could be viewed as a type of negative sampling. The authors provide a response, arguing that GCL can incorporate neighboring nodes as a type of view and that negative samples are not necessary for preventing representation collapse in PROP. This discussion highlights the need for further clarification on the boundaries between different approaches to graph learning.

Overall, this work contributes useful insights into the mechanisms underlying Graph Contrastive Learning and proposes a new framework that warrants further exploration. While some questions and concerns remain, the authors' responses and acknowledgments of limitations demonstrate a clear understanding of the challenges and opportunities. I believe the paper needs a little bit more work before publication.

**Additional Comments On Reviewer Discussion:**

Reviewers were engaged. They like the idea but there are too many drawbacks right now. The paper needs a bit more work.

---

### Decision · Program_Chairs · 2025-01-22

Reject